# LeVo: High-Quality Song Generation with Multi-Preference Alignment

**Shun Lei**[1†], **Yaoxun Xu**[1†], **Zhiwei Lin**[1], **Huaicheng Zhang**[3], **Wei Tan**[2*], **Hangting Chen**[2*],
**Yixuan Zhang**[2], **Chenyu Yang**[4], **Haina Zhu**[5], **Shuai Wang**[6], **Zhiyong Wu**[1*], **Dong Yu**[2*]

[1] Shenzhen International Graduate School, Tsinghua University, Shenzhen
[2] Tencent AI Lab [3] Wuhan University
[4] The Chinese University of Hong Kong, Shenzhen (CUHK-Shenzhen), Shenzhen, China
[5] X-LANCE Lab, Shanghai Jiao Tong University, Shanghai
[6] School of Intelligence Science and Technology, Nanjing University, Suzhou, China
`{leis21, xuyx22}@mails.tsinghua.edu.cn, erichtchen@tencent.com,`
`zywu@sz.tsinghua.edu.cn`

## Abstract

Recent advances in large language models (LLMs) and audio language models have significantly improved music generation, particularly in lyrics-to-song generation. However, existing approaches still struggle with the complex composition of songs and the scarcity of high-quality data, leading to limitations in audio quality, musicality, instruction following, and vocal-instrument harmony. To address these challenges, we introduce LeVo, a language model based framework consisting of LeLM and Music Codec. LeLM is capable of parallel modeling of two types of tokens: mixed tokens, which represent the combined audio of vocals and accompaniment to achieve better vocal-instrument harmony, and dual-track tokens, which separately encode vocals and accompaniment for high-quality song generation. It employs two decoder-only transformers and a modular extension training strategy to prevent interference between different token types. To further enhance musicality and instruction following ability, we introduce a multi-preference alignment method based on Direct Preference Optimization (DPO). This method handles diverse human preferences through a semi-automatic data construction process and post-training. Experimental results demonstrate that LeVo significantly outperforms existing open-source methods in both objective and subjective metrics, while performing competitively with industry systems. Ablation studies further justify the effectiveness of our designs. Audio examples and source code are available at https://levo-demo.github.io/ and https://github.com/tencent-ailab/songgeneration.

## 1 Introduction

Songs represent one of the most significant forms of musical expression, encapsulating human creativity and intelligence. By blending expressive vocals with a rich tapestry of instruments, songs convey the emotions and thoughts of their creators, offering artistic appeal and broad cultural influence. Rapid advances in Artificial Intelligence Generated Content (AIGC) have already transformed creative domains—from text generation [1–3] and speech synthesis [4–7] to more sophisticated domains such as image generation [8–10] and instrumental music generation [11–14]. Nevertheless, creating songs remains a complex challenge for AIGC systems: they must generate high-quality vocal and instrumental tracks and integrate them seamlessly while maintaining musicality and

---

*Corresponding author, [†] Equal contribution.

39th Conference on Neural Information Processing Systems (NeurIPS 2025).

following instructions. These challenges are amplified for long-form song generation, where modeling complexity and computational cost increase significantly.

Early song generation approaches treat the combined audio of the vocals and accompaniment as a single prediction target. Jukebox [15] introduced the idea of using language models (LMs) to predict discrete codes ("mixed tokens") extracted from the song audio. Later studies, such as SongCreator [16] and MusiCot [17], have further optimized the language model to improve musicality. Despite their success, mixed token-based approaches exhibit a critical limitation: the restricted vocabulary cannot fully capture the intricate combination of vocals and accompaniment, which leads to lower sound quality. YuE [18] tackles this limitation by introducing a dual-track token prediction strategy that generates separate token sequences for the vocal and instrumental tracks. SongGen [19] extends this idea and shows that interleaving the two streams along the temporal dimension, instead of predicting them in parallel, reduces interference between token types and achieves better performance. While these methods produce high-quality vocals and accompaniments, they struggle to maintain vocal-instrument harmony due to the independent prediction. The interleaved prediction mode dramatically increases the length of the token sequence, which limits scalability and makes it difficult to maintain musicality and follow instructions.

Moreover, available datasets in the song generation community have long been constrained by highly uneven quality and unreliable music annotations, which exhibit several challenges. On the one hand, models trained on such noisy data are generic enough to model song signals, but they lack prior knowledge of musicality to bias the generated songs toward listener-preferred music. On the other hand, unreliable annotations limit the model's ability to follow instructions such as lyrics and prompts. Similar issues in text generation are typically circumvented by using the reinforcement learning (RL) method (e.g., RLHF), but music annotation requires specialized domain knowledge and evaluation along multiple perceptual dimensions.

This paper presents LeVo, a language model based song generation framework consisting of LeLM and Music Codec. LeLM is used to model mixed tokens and dual-track tokens in parallel. Mixed tokens guide the overall arrangement of melody, rhythm, and tempo, ensuring vocal-instrument harmony, while dual-track tokens capture finer nuances to improve the sound quality and musicality. The Music Codec then reconstructs the dual-track tokens into high-fidelity music audio. To prevent interference between these two token types, the proposed LeLM utilizes a language model with an autoregressive (AR) decoder and a modular extension training strategy. This framework allows flexible control over musical elements, enabling text-based descriptions to influence instrumental, genre, and emotion, and audio prompts to achieve zero-shot style transfer. Additionally, to address the challenges of musicality and instruction following, we further introduce a DPO-based [20] multi-preference alignment method to improve the model's performance across multiple dimensions. The main contributions of this paper are summarized as follows:

- LeVo presents a new paradigm to optimize LM-based music generation by three-stage training: pre-training, modular extension training, and multi-preference alignment. Pre-training brings generation diversity and vocal-instrument harmony, modular extension training improves sound quality and musicality without disrupting the pre-trained knowledge, and multi-preference alignment further enhances instruction following and musicality.

- The architecture of LeLM is carefully designed using a language model with an AR decoder to predict mixed and dual-track tokens in parallel. Compared to single token prediction (only using mixed or dual-track tokens) or straightforward joint training, our method simultaneously optimizes musicality, vocal–instrument harmony, and sound quality.

- To balance the multi-dimensional demands of song generation, we propose a multi-preference alignment method based on DPO, which is the first attempt to apply multi-preference DPO to song generation. Our strategy handles diverse human preferences by corresponding semi-automatic data construction processes and DPO fine-tuning. The final combined model achieves higher musicality and better instruction following.

In terms of musicality, sound quality, vocal–instrument harmony, and instruction following, the proposed LeVo model significantly improves over the open-source music generation models and performs competitively with current state-of-the-art industry systems.

## 2 Related Work

**Music Generation**    Early efforts [21, 22] in music generation primarily focused on symbolic music, which is limited by fixed instruments and lacks expressiveness. Recently, LLMs have demonstrated strong reasoning capabilities and scaling behaviors [1–3]. Several works [11, 14, 23] have achieved the end-to-end music generation using language models. These works encode music as discrete token sequences by the Vector Quantised-Variational Autoencoder (VQ-VAE) [24] or the Residual Vector Quantization (RVQ) [25, 26], and then process them with language models. However, their performance was constrained by quantization loss. To address this, some studies adopt diffusion models, which have been proven effective at modeling continuous representations [27, 28]. However, these methods face challenges in long-context generation tasks due to the significantly increased computational demands as the sequence lengthens and the model size grows. Recently, MeLoDy [12] and AudioLDM 2 [29] combined language models and diffusion models, achieving state-of-the-art (SOTA) performances with high-fidelity and musicality. We extend this idea by using a diffusion model as the decoder of the codec and LeLM, which considers the complex components of songs, to better achieve song generation.

Furthermore, several intriguing studies have focused on applying RL in music generation. BATON [30] integrates a reward model, which is trained based on human preferences, into the standard diffusion loss to guide the training of diffusion models. MusicRL [31] fine-tunes MusicLM [14] using reward functions derived from large-scale user feedback through Reinforcement Learning from Human Feedback (RLHF) to learn human preferences. Tango2 [32] semi-automatically generates preference datasets for DPO [20] training, providing a cheaper and more robust alternative. Different from these works that only focus on one specific dimension in non-vocal music generation, to our knowledge, we are the first attempt to align multiple preferences in song generation.

**Song Generation**    Recently, several studies have explored the challenging task of song generation, which involves generating vocals with rich accompaniment. Jukebox [15] is one of the pioneering efforts, which utilizes a multi-scale VQ-VAE to compress songs into discrete codes and models them with a cascaded transformer model. To improve the musicality of generated songs, SongCreator [16] introduced a dual-sequence language model to capture the relationship between vocals and accompaniment. MusiCot [17] predicts coarse-grained style representations to first sketch an overall musical structure before generating audio tokens, thereby improving the coherence and creativity of the generated songs. However, these approaches are affected by limited vocabulary and interference between vocals and accompaniment. To address these issues, Melodist [33] and MelodyLM [34] utilize a multi-stage process to sequentially produce vocals and accompaniment. YUE [18] and SongGen [19] attempt to operate on sequences consisting of vocal tokens and accompaniment tokens. Different from these, DiffRhythm [35] uses a diffusion-based approach to generate full-length songs. Despite the advancements in these methods, they still struggle with the complex composition of songs and the uneven quality of data, which makes it difficult to maintain musicality, lyrics alignment, and vocal-instrument harmony. Industry tools such as Meruka [36], Seed-Music [37], Suno [38], and Udio [39] have demonstrated promising results for song generation, but none have fully disclosed their technical details. Our proposed LeVo outperforms previous academic methods by predicting mixed and dual-track tokens in parallel while explicitly preventing their mutual interference. Furthermore, by leveraging a DPO-based multi-preference alignment method, LeVo achieves additional improvements, yielding human-perceptual quality that is competitive with leading industry tools.

## 3 Method

### 3.1 Overview

Let $\mathbf{x} \in \mathcal{X}$ represent a song audio. A song generation process can be defined as $f : \mathcal{C} \mapsto \mathcal{X}$, where $\mathcal{C}$ is the set of conditions. However, end-to-end generating a high-fidelity song $\mathbf{x}$ from $\mathbf{C}$ with a neural network $f$ remains challenging to date. In the same spirit as previous works [16, 18], we introduce a language-alike token sequence, denoted as $\mathbf{S} = (S_1, \ldots, S_N)$, to capture significant structural information within the song and to embody language models as the "brain" for creating songs.

Our proposed LeVo generates songs from lyrics and can be supplemented with text descriptions and audio prompts. As illustrated in Figure 1, it consists of LeLM and Music Codec. The encoder

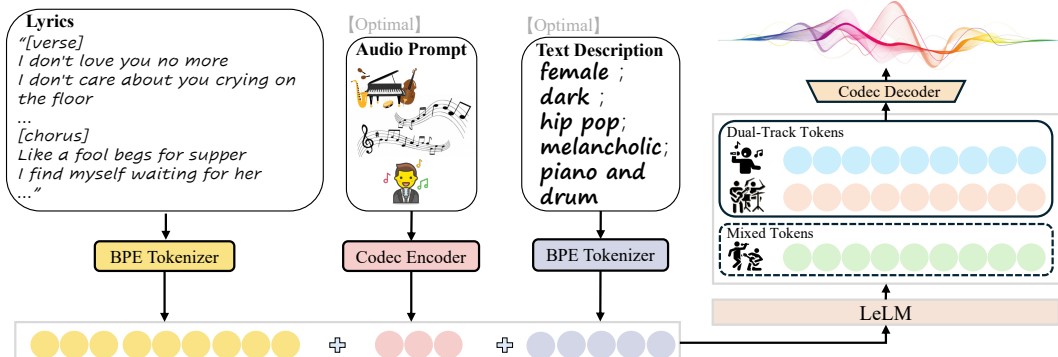

Figure 1: The overview of LeVo, a song generation framework based on lyrics, optional text descriptions, and optional audio prompts. It consists of LeLM and Music Codec.

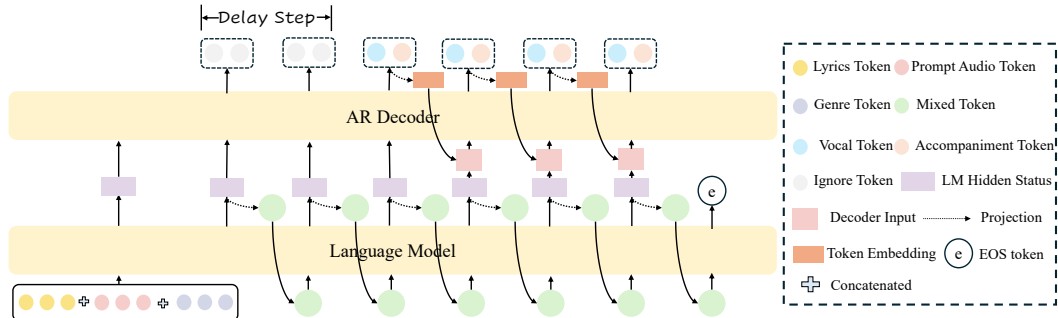

Figure 2: The architecture of LeLM, which consists of a language model and an AR decoder.

of Music Codec is utilized to extract mixed tokens (i.e., $\mathbf{S}_m \in \mathcal{S}_m$), vocal tokens (i.e., $\mathbf{S}_v \in \mathcal{S}_v$), and accompaniment tokens (i.e., $\mathbf{S}_a \in \mathcal{S}_a$) from the song audio as prediction targets for LeLM. To parallelly predict the mixed tokens and dual-track tokens given $\mathbf{C}$, we propose a language model with an AR decoder named LeLM, as illustrated in Figure 2. The lyrics, optional text descriptions, and optional audio prompts are concatenated and fed into the LeLM as prefix context. Then, the decoder of Music Codec is used to generate high-quality and high-fidelity song audio based on dual-track tokens that contain more detailed information about the vocal and accompaniment tracks.

Furthermore, to address the challenges posed by interference between different types of tokens and issues such as musicality and instruction following, we propose a three-stage training strategy and a DPO-based multi-preference alignment method. In the remainder of this section, we present the LeVo architecture details and the training process described above.

## 3.2 Language Modeling of LeVo

Although modeling dual-track tokens struggles to maintain vocal-instrument harmony, its significant advantages in sound quality have led to widespread attempts [18, 19]. The interleaving pattern, in particular, has shown improved performance over the parallel pattern by reducing interference between different token types. However, limited by the substantial increase in sequence length, the interleaving pattern struggles to scale to long-context song generation. To address this, LeVo introduces a novel parallel modeling approach that avoids cross-token interference without significantly lengthening the sequence, thereby improving both vocal-instrument harmony and sound quality.

As shown in Figure 2, this approach consists of a language model and an autoregressive (AR) decoder. The language model utilizes a decoder-only Transformer architecture, which has been widely adopted in previous works on music generation [12, 14, 18]. It focuses on the next-token prediction task for mixed tokens to capture the high-level structural information about the song, such as melody, rhythm,

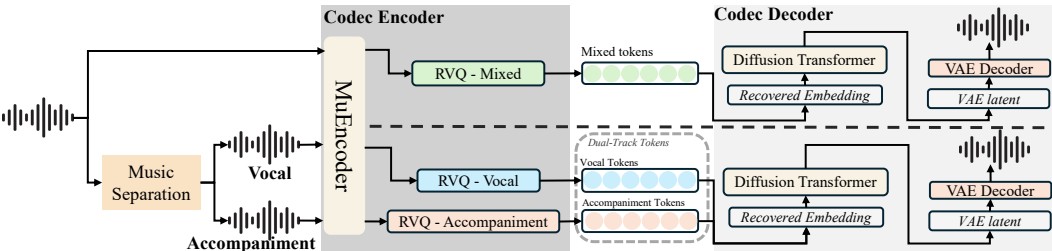

Figure 3: The framework of the Music Codec in LeVo.

and tempo, ensuring vocal-instrument harmony. This task can be formulated as:

$$p(\mathbf{S}_m|\mathbf{C};\boldsymbol{\theta}) = \prod_{t=0}^{T} p(\mathbf{S}_{m,t}|\mathbf{S}_{m,<t},\mathbf{C};\boldsymbol{\theta}) \quad (1)$$

where $\mathbf{C}$ denotes conditions, including lyrics, optional text descriptions, and optional audio prompts.

To facilitate high-quality song generation, we use an AR decoder that predicts dual-track tokens in parallel based on the outputs of the language model. It also utilizes a decoder-only Transformer architecture with significantly fewer layers and parameters than the language model and is designed to refine the information generated by the language model through modeling of finer acoustic details. As the hidden states of the language model contain richer semantic and acoustic information than the mixed tokens, we pass these hidden states to the AR decoder. Specifically, at each time step, the embeddings of the predicted vocal and accompaniment tokens from the previous time step, along with the hidden states of the language model's last layer, are concatenated to form the input of the AR decoder. Two groups of linear heads are then employed to predict the audio tokens for each track based on the output of the AR decoder. Moreover, to provide rich contextual information, we introduce a delay pattern between the dual-track tokens and mixed tokens. This means that when predicting the dual-track tokens at the $t$-th time step, the model can consider not only the language model's output at time steps less than or equal to $t$, but also the output for the next $k$ time steps, where $k$ represents the number of time steps for the delay. This process can be simply represented as:

$$p(\mathbf{S}_v, \mathbf{S}_a|\mathbf{C};\boldsymbol{\theta}) = \prod_{t=0}^{T-k} p(\mathbf{S}_{v,t}, \mathbf{S}_{a,t}|\mathbf{S}_{v,<t}, \mathbf{S}_{a,<t}, \mathbf{S}_{m,<t+k}, \mathbf{C};\boldsymbol{\theta})$$
$$\prod_{t=T-k+1}^{T} p(\mathbf{S}_{v,t}, \mathbf{S}_{a,t}|\mathbf{S}_{v,<t}, \mathbf{S}_{a,<t}, \mathbf{S}_m, \mathbf{C};\boldsymbol{\theta}) \quad (2)$$

### 3.3 Music Codec in LeVo

Building upon MuCodec [40]—an efficient 48kHz music codec capable of operating at low bitrates—we have developed the Music Codec in LeVo, as shown in Figure 3. The codec consists of an encoder and a decoder. The encoder is composed of the MuEncoder [40] and an RVQ: the MuEncoder extracts music-relevant representations, while the RVQ discretizes these representations into tokens. As the central intermediary in LeVo, these tokens serve not only as the prediction targets for LeLM but also play a pivotal role in achieving high-fidelity music reconstruction by acting as inputs for the decoder. The codec decoder comprises a diffusion transformer and a Variational Autoencoder (VAE) decoder. The diffusion transformer reconstructs VAE features from the token-derived embeddings, which the VAE decoder then converts directly into audio. This approach is significantly faster than methods that rely on Mel-spectrograms as intermediate steps for audio reconstruction.

Music inherently exhibits a rich hierarchical structure, particularly in the complex interplay between vocals and accompaniment. To generate expressive vocals alongside clear and diverse accompaniment, we propose two strategies: the mixed token approach and the dual-track token approach. In the mixed token approach, music is processed by the encoder to extract mixed tokens. These tokens encapsulate information from both the vocals and the accompaniment. The decoder then uses the recovered embeddings from these mixed tokens to reconstruct the VAE latent, ultimately restoring the original music. In the dual-track token approach, a pretrained music separation model is employed to separate the vocal and accompaniment tracks. Each track is then processed by its respective encoder to produce

vocal tokens and accompaniment tokens. The decoder conditions on the recovered embeddings from both sets of tokens to reconstruct the VAE latent of the original music, and finally restores the music.

## 3.4 DPO-based Multi-Preference Alignment

The scarcity of high-quality data has severely constrained the effectiveness of previous works in academic research. On the one hand, uneven data quality prevents models from learning prior knowledge about musicality. On the other hand, the lack of accurate music annotations, such as lyrics and descriptive text, undermines the model's ability to perform lyrics alignment and prompt-driven control. To address these challenges, we first adopt a semi-automatic multi-dimensional preference data construction method to generate large-scale preference data at a low cost for DPO fine-tuning.

First, we generate 20,000 lyrics by using a large language model. For each lyric, we randomly select an audio prompt from a pre-defined dataset that includes text descriptions. Then, we generate multiple samples under three different conditions: using only the text description, using only the audio prompt, and using both the text description and the audio prompt. For the three dimensions of lyrics alignment, prompt consistency, and musicality, we then introduce three distinct strategies for constructing preference data corresponding to each dimension.

**Strategy 1: Lyric Alignment Preferences** The lyrics of the generated audio might exhibit misalignment with the input text, for example, word substitution, insertion, and deletion. To eliminate such a misalignment, we construct win-lose pairs using phoneme errors calculated by automatic speech recognition (ASR). Two audio samples will form a win-lose pair if their phoneme error numbers are larger than 40. We pick out all pairs satisfying the phoneme error gap condition.

**Strategy 2: Prompt Consistency Preferences** Considering the noise observed in the annotation of the training data, the style of the generated audio might be inconsistent with the input prompt. To improve the ability to prompt-driven control, we construct preference pairs using similarity scores calculated by the MuQ-MuLan [41] model. We aim to ensure that the winning audio sample is both strongly aligned with the prompt and substantially better aligned than the losing sample. For text descriptions, the winning sample must achieve a score of at least 0.3, exceeding the losing sample's score by at least 0.1. For audio prompts, the winning sample must reach a score of at least 0.75, again with a score difference greater than 0.1 compared to the losing sample.

**Strategy 3: Musicality Preferences** To enhance musicality, we employ a three-stage procedure to sift through win-lose pairs. Initially, we carry out crowdsourcing to rank a subset of the generated samples, which included 4,000 different sets of lyrics. We select those with high levels of agreement among human evaluators, forming a dataset of win-lose pairs. In the second stage, we train a reward model based on the datasets curated from the first stage. Finally, in the third stage, we deploy the reward model across all generated samples. We establish a threshold to filter out data that exhibits a large gap in reward scores and tune the threshold to ensure that the selected win-lose pairs have an accuracy above 80%. Through this three-stage procedure, we ultimately collect roughly 60,000 win–lose pairs. The rationale behind this procedure is that while the size of the human-labeled dataset is limited, the use of a reward model enables us to identify all potential pairs.

The above paired data are then used for DPO fine-tuning to address the challenges posed by scarce high-quality data in the dimensions of lyrics alignment, prompt consistency, and musicality. To balance different aspects of music generation, we introduce an interpolation-based multi-preference alignment method inspired by DNI (Deep Network Interpolation) [42]. In this mode, we fine-tune the model separately on the preference data curated for each strategy to obtain the network parameters dedicated to each preference. Then, we apply linear interpolation across all parameters of these three specialized networks to produce the final model. Moreover, this approach also supports smooth transitions in performance according to specific demands by the controllable interpolation coefficient.

## 3.5 Three-Stage Training Paradigm

To facilitate the performance of LM-based song generation approaches, we introduce a novel three-stage training paradigm that progressively enhances the capabilities of LeLM:

Table 1: Objective results of comparison and ablation systems for song generation. The asterisk (*) denotes that we reproduce SongGen using our training data. The overall first and second results are marked with **bold** and underline, respectively.

| Models | FAD ↓ | MuQ-T ↑ | MuQ-A ↑ | PER ↓ | Content Scores ↑ | | | |
|---|---|---|---|---|---|---|---|---|
| | | | | | CE | CU | PC | PQ |
| Suno-V4.5 | 2.59 | **0.34** | 0.84 | 21.6 | 7.65 | 7.86 | 5.94 | 8.35 |
| Haimian | 2.97 | 0.22 | − | 11.8 | 7.56 | 7.85 | 5.89 | 8.27 |
| Mureka-O1 | **2.50** | 0.33 | **0.87** | **7.2** | 7.71 | 7.83 | **6.39** | 8.44 |
| YuE | 2.65 | 0.27 | 0.74 | 36.4 | 7.13 | 7.39 | 5.90 | 7.77 |
| DiffRhythm | 4.86 | 0.26 | 0.51 | 12.3 | 6.65 | 7.32 | 5.71 | 7.77 |
| ACE-Step | 2.69 | 0.28 | − | 37.1 | 7.37 | 7.52 | 6.26 | 7.85 |
| SongGen* | 2.68 | 0.25 | 0.80 | 27.5 | 7.63 | 7.79 | 5.94 | 8.37 |
| LeVo | 2.68 | **0.34** | 0.83 | **7.2** | **7.78** | **7.90** | 6.03 | **8.46** |
| w/o Train stage 2 | 2.71 | 0.28 | 0.82 | 17.5 | 7.76 | 7.81 | 5.69 | 8.44 |
| w/o AR decoder | 2.83 | 0.27 | 0.80 | 26.0 | 7.54 | 7.71 | 5.61 | 8.32 |
| w/o Dual-track | 2.83 | 0.33 | 0.83 | 11.0 | 7.72 | 7.88 | 5.82 | 8.43 |
| w/o DPO | 2.60 | 0.31 | 0.82 | 10.6 | 7.70 | 7.86 | 5.89 | 8.39 |

**Stage 1: Pre-training**    The language model in LeLM is first trained on large-scale music data to align the modalities between the various conditioning inputs and the mixed tokens. To ensure the model functions under different input conditions, both audio prompts and text descriptions undergo 50% random drop. At this stage, the AR decoder is frozen to facilitate the language model to focus solely on mixed tokens, bringing generation diversity and vocal-instrument harmony.

**Stage 2: Modular Extension Training**    Building on Stage 1, we further train the AR decoder in LeLM to model dual-track tokens in parallel. This captures fine-grained acoustic details and nuances, thereby improving sound quality and musicality. To preserve the pre-trained knowledge, all modules trained in Stage 1 are frozen.

**Stage 3: Multi-preference alignment**    Finally, we employ the semi-automatic data construction processes to create a multi-dimensional preference dataset and fine-tune the entire LeLM using the DPO loss. This stage significantly enhances musicality and the ability to follow instructions.

## 4    Experiments

### 4.1    Experimental setup

**Dataset**    LeVo is trained on a large-scale music dataset comprising 2 million songs (approximately 110,000 hours). We adopt an automatic data processing pipeline similar to AutoPrep [43] to address the challenge of the lack of accurate annotations. Initially, we use the Demucs [44, 45] to extract the vocals from the songs. Whisper [46] and wav2vec 2.0 [47, 48] are then leveraged for lyric recognition and to provide timestamps for the lyrics. To enable the model to learn the structure of the songs, we adopt the All-In-One model [49] to automatically extract music segments (e.g., verse, chorus, etc.). Additionally, we annotate all tracks utilizing Qwen2-Audio [50] to obtain open-vocabulary tags.

**Model setup**    Our proposed LeLM has approximately 2B parameters. For the MuEncoder model that generates mixed tokens and dual-track tokens, we utilize the pre-trained weights from Mucodec [40] with 300M parameters and a frame rate of 25HZ. Our diffusion model features a model size of about 700M parameters to convert tokens into high-quality waveforms. We replicate the VAE used in Stable Audio with 150M parameters. Additionally, for lyrics and text description, LeVo employs a byte pair encoding (BPE)-based Qwen2 tokenizer [51] to process raw text. Detailed configurations and training setups can be found in the Appendix B.

**Evaluations**    For objective evaluation, both the Fréchet Audio Distance (FAD) [52] and the Phoneme Error Rate (PER) are employed. To calculate PER, the vocal track is first extracted by Demucs, and then Whisperlarge-v2 is utilized for lyric recognition. We employ MuQ-MuLan [41], a contrastive

Table 2: Subjective results of comparison and ablation systems for song generation. The asterisk ($*$) denotes that we reproduce SongGen using our training data. The overall first and second results are marked with **bold** and underline, respectively.

| Models | MOS ↑ | | | | | |
|---|---|---|---|---|---|---|
| | **OVL** | **MEL** | **HAM** | **SSC** | **AQ** | **LYC** |
| Suno-V4.5 | **3.59** | **4.10** | **3.93** | **4.19** | **4.00** | 3.17 |
| Haimian | 3.05 | 3.51 | 3.55 | 3.62 | 3.87 | 3.32 |
| Mureka-O1 | 3.42 | 3.88 | 3.89 | 4.14 | 3.87 | 3.32 |
| YuE | 2.45 | 3.04 | 2.94 | 3.53 | 3.08 | 2.41 |
| DiffRhythm | 2.60 | 3.18 | 3.22 | 3.55 | 3.09 | 2.69 |
| ACE-Step | 2.26 | 3.02 | 3.30 | 3.21 | 2.36 | 2.22 |
| SongGen$^*$ | 2.91 | 3.43 | 3.44 | 3.66 | 3.69 | 2.84 |
| LeVo | 3.42 | 3.93 | 3.90 | 4.09 | 3.96 | **3.38** |
|    w/o Train stage 2 | 3.29 | 3.76 | 3.77 | 3.80 | 3.96 | 2.91 |
|    w/o AR decoder | 2.93 | 3.44 | 3.34 | 3.59 | 3.71 | 2.74 |
|    w/o Dual-track | 3.25 | 3.82 | 3.84 | 3.96 | 3.86 | 3.18 |
|    w/o DPO | 3.18 | 3.71 | 3.76 | 3.97 | 3.93 | 3.18 |

music-language pre-training model, to measure the similarity between the generated song and its text description (MuQ-T) or audio prompt (MuQ-A). Additionally, the Meta Audiobox-Aesthetic [53] is used to capture perceived musical aesthetics, including content enjoyment (CE), content usefulness (CU), production complexity (PC), and production quality (PQ). For subjective evaluation, we conduct a mean opinion score (MOS) listening test. Specifically, we enlist ten music professionals to rate each generated song on a scale from 1 to 5 across six aspects: overall quality (OVL), focusing on musicality and naturalness; vocal melodic attractiveness (MEL); vocal-instrument harmony (HAM); song structure clarity (SSC); audio sound quality (AQ); lyrics following accuracy (LYC). None of the evaluators participated in model training to ensure objectivity. During the experiments, we asked participants to listen to the entire music generated by different models with the same input before scoring, thereby obtaining scores that reflect the relative differences between models. The Appendix D shows details.

**Comparison systems**   We conducted a comprehensive comparison between LeVo and multiple systems. We selected three leading industry systems for benchmarking: Suno V4.5 [38], Mureka-O1 [36], and Haimian [54]. It is important to note that due to the black-box nature of these closed-source models, our evaluation conducted in May 2025 reflects the performance of these systems at that specific time. Furthermore, we selected four open-source academic systems for benchmarking: YuE[2] [18], DiffRhythm[3] [35], ACE-Step[4] [55] and SongGen [19]. Notably, the official SongGen model can only generate songs up to 30 seconds with a sampling rate of 16 kHz. For fair comparison, we reproduced its Interleaving (A-V) mode based on our dataset and codec.

## 4.2   Comparison with the State-of-The-Art Systems

**Objective results**   Table 1 summarizes the objective results. LeVo achieves either superior or competitive performance across all metrics. It obtains the highest MuQ-T scores (0.34) and lowers PER (7.2%) among all systems, and the highest MuQ-A scores (0.83) after closed-source systems, which demonstrates its powerful instruction following ability, especially in text description and lyrics. LeVo also records the highest scores on the Audiobox-Aesthetic dimensions of CE (7.78), CU (7.90), and PQ (8.46), indicating its superior perceived musicality. For FAD and PC, LeVo achieves comparable results to other open-source approaches that focus on dual-track tokens modeling or diffusion-based generation, reflecting comparable sound quality, whereas industry model Mureka-O1 records the lowest FAD and higher PC, attributed to its exceptional sound quality.

---

[2]YuE is tested using the model released at https://github.com/multimodal-art-projection/YuE

[3]DiffRhythm is tested using DiffRhythm-full released at https://github.com/ASLP-lab/DiffRhythm

[4]ACE-Step is tested using the text2music demo space at https://huggingface.co/spaces/ACE-Step/ACE-Step

Table 3: Comparison of various DPO methods across multiple objective metrics. The overall first and second results are marked with **bold** and underline, respectively.

| Models | FAD ↓ | MuQ-T ↑ | MuQ-A ↑ | PER ↓ | Content Scores ↑ | | | |
|---|---|---|---|---|---|---|---|---|
| | | | | | CE | CU | PC | PQs |
| w/o DPO | **2.60** | 0.31 | 0.82 | 10.6 | 7.70 | 7.86 | 5.89 | 8.39 |
| with Strategy 1 | 2.85 | 0.30 | 0.81 | **6.5** | 7.72 | 7.86 | 5.97 | 8.42 |
| with Strategy 2 | 2.89 | **0.34** | **0.83** | 10.3 | 7.75 | 7.87 | 5.96 | 8.43 |
| with Strategy 3 | 2.63 | 0.32 | 0.82 | 11.2 | **7.78** | **7.93** | **6.16** | 8.45 |
| Mixed Training | 2.75 | 0.33 | **0.83** | 7.5 | 7.76 | 7.89 | 6.04 | 8.43 |
| LeVo (Interpolation) | 2.68 | **0.34** | **0.83** | 7.2 | **7.78** | 7.90 | 6.03 | **8.46** |

**Subjective results**    As shown in Table 2, LeVo outperforms all open-source academic approaches on all dimensions, highlighting the musicality, audio quality, vocal-instrument harmony, and lyrics alignment of the generated song. It is important to note that our proposed parallel prediction method surpasses the interleaved pattern of SongGen, whose performance degrades on long-context song generation. Although the industry system Suno-V4.5 excels in most metrics, LeVo exceeds it by 0.21 points on LYC, which demonstrates LeVo's excellent lyric-alignment capability. Furthermore, LeVo achieves MOS scores close to Suno-V4.5 on OVL, MEL, HAM, and AQ, and outperforms all other systems, indicating its superior overall quality—especially in musicality, vocal-instrument harmony, and sound quality. However, Suno-V4.5 and Mureka-O1 show slightly better song structure clarity scores, which suggests room for improvement in music structural modeling.

## 4.3   Ablation study

**Framework**    As presented in the lower part of Table 1 and 2, we demonstrate the effectiveness of several techniques used in LeVo, including the modular extension training strategy, the AR decoder, and the prediction of dual-track tokens. It is observed that jointly training the language model and AR decoder ("w/o train stage 2") decreases all metrics, particularly in PER, LYC, SSC, and MEL. Removing the AR decoder further amplifies this decrease. The results indicate that the proposed AR decoder and modular extension training strategy are crucial for preventing interference between mixed and dual-track tokens, thereby improving musicality and intelligibility while maintaining vocal-instrument harmony. Moreover, we find that removing the prediction of dual-track tokens causes a significant decline in FAD, PER, and PC (objective) together with OVL and LYC (subjective), whereas MEL, SSC, and AQ decline more modestly. It confirms that the prediction of dual-track tokens is essential for sound quality and intelligibility, and also has a few impact on musicality. We further explore the reconstruction performance of dual-track tokens in Appendix C.

**DPO-based Multi-Preference Alignment**    As shown in the lower part of Table 1 and Table 2, removing DPO leads to significant decreases in PER, OVL, MEL, and LYC, indicating that DPO-based multi-preference alignment effectively enhances both musicality and lyric alignment. We also conducted a MOS listening test to evaluate the alignment between the given prompt and the generated music. In this experiment, "LeVo" achieved a score of 3.175, while "LeVo w/o DPO" scored 3.055, suggesting that DPO-based multi-preference alignment also improves prompt consistency.

Table 3 further details how different preferences affect objective performance. When the model is fine-tuned with a single preference, each strategy selectively enhances its targeted capability: Lyric Alignment Preference (Strategy 1) reduces PER from $10.6\%$ to $7.0\%$; Prompt Consistency Preference (Strategy 2) yields the highest MuQ-T (0.34) and MuQ-A (0.83) scores; and Musicality Preference (Strategy 3) achieves the best Audiobox-Aesthetic results. We then compare our proposed interpolation-based multi-preference alignment method with a simple mixed training approach, where preference data from all strategies are combined during post-training. While mixed training improves both instruction following and musicality compared to the no-DPO baseline—albeit with a slight increase in FAD—our proposed interpolation method outperforms mixed training across most metrics, demonstrating a better balance among the multi-dimensional demands of music generation. We further analyze the smooth performance transitions enabled by the controllable interpolation coefficient in Appendix G.

# 5    Conclusion and Discussion

**Conclusion**    In this paper, we present LeVo, a song generation model that parallelly predicts mixed tokens for vocal-instrument harmony and dual-track tokens for high-fidelity vocals and accompaniment. By leveraging the LeLM and its training strategy, LeVo eliminates the mutual influence between different types of tokens. In addition, we propose a multi-preference alignment method to improve the performance in terms of lyrics alignment, prompt consistency, and musicality. Experimental results demonstrate the excellent song generation capabilities of LeVo.

**Limitations**    Despite LeVo outperforming all academic approaches, its audio quality remains constrained by the variable quality of training data and the use of discrete tokens, leaving a gap from state-of-the-art industry models. Moreover, there is still a disparity between LeVo and Suno across several subjective metrics, suggesting that further alignment with human preference is necessary. In addition, the scarcity and high cost of annotated music data have led us to rely heavily on pre-trained models for pseudo-labeling, which inevitably impacts LeVo's instruction following capability. On one hand, since text descriptions are generated using Qwen2-Audio, the diversity and richness of effective prompts are limited. On the other hand, errors accumulated in lyric extraction, structure recognition, and music captioning processes introduce noise into supervision, reducing the fidelity of instruction alignment. These factors collectively contribute to the remaining gap between LeVo and top-tier proprietary systems in both controllability and overall quality.

**Broader Impact**    The proposed work has the potential to empower both content creators and novices to express their musical creativity with a low entry barrier, while also streamlining the workflow of experienced music producers. However, given LeVo's ability to perform style transfer and end-to-end song generation, it could be misused for creating misinformation, deepfake audio, or other harmful content. To promote responsible deployment, we plan to establish appropriate safeguards and usage guidelines when releasing the open-source code and model checkpoints. In addition, we examine the risks of training data memorization and content leakage, finding no evidence of direct reproduction from the training set under identical inputs (see Appendix H for detailed analysis).

## Acknowledgements

This work is supported by National Natural Science Foundation of China (62076144) and Shenzhen Science and Technology Program (JCYJ20220818101014030).

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

# A    Project Contributors

- Project Sponsors: Dong Yu, Zhiyong Wu
- Core Contributors:
    - Data Collection & Washing: Wei Tan, Hangting Chen, Jianwei Yu
    - Music Encoder: Shun Lei, Hangting Chen, Haina Zhu
    - Diffusion & VAE: Yaoxun Xu, Zhiwei Lin, Hangting Chen
    - LLM Pre-train & Post-train: Shun Lei, Wei Tan, Huaicheng Zhang, Chenyu Yang, Hangting Chen, Jianwei Yu
    - Music Caption: Yixuan Zhang, Wei Tan
- Other Contributors: Chuan Lin, Rongzhi Gu*, Ruoning Hong, Shang Gao, Shulan Zhang, Yi Luo*, Yizhi Zhou* (∗ contributed while working in Tencent AI Lab)

# B    Training and Implementation Details

## B.1    LeLM

Our LeLM consists of a language model and an AR decoder. The language model is a 28-layer Transformer with 1536 hidden size. The AR decoder is a 12-layer Transformer with 1536 hidden size. We provide detailed hyperparameter settings about this model configuration in Table 4. We collected approximately 110,000 hours of songs from the internet for model training, comprising part of the DISCO-10M [56], Million Song Dataset [57], and some copyrighted in-house data.

During training, we used 32 NVIDIA H20 GPUs, with a batch size of 2 for each GPU to train the LeLM for 265K steps. Of these, 200k steps are required for the pre-training, with 60k steps for modular extension training, then 5k steps for multi-preference alignment. Adam optimizer is used with $\beta_1 = 0.9, \beta_2 = 0.98, \epsilon = 10^{-9}$ and follow the same learning rate schedule in [58]. Consistently, top-$k$ sampling is adopted for inference, in which $k$ and temperature are set to 50 and 0.9, respectively.

Table 4: Hyper-parameters of LeLM.

|  | Hyper-parameter | Value |
|---|---|---|
| Language model | Encoder Layers | 28 |
|  | Hidden Size | 1536 |
|  | Attention Head | 12 |
|  | Feed-Forward Dim | 8960 |
|  | Max Context Length (in #tokens) | 8196 |
| AR Decoder | Decoder Layers | 12 |
|  | Hidden Size | 1536 |
|  | Attention Head | 12 |
|  | Feed-Forward Dim | 8960 |
|  | Max Context Length (in #tokens) | 10000 |
| Total Number of Parameters |  | 2136M |

## B.2    Music Codec

The Music Codec consists of four main components. First, the MuEncoder [40] extracts representations specifically designed for music. Next, the RVQ module discretizes these representations. The Diffusion Transformer (DiT) then transforms the discretized representations into Variational AutoEncoder (VAE) representations, which are finally decoded back into waveforms by the VAE. In the following parts, we will provide a detailed description of the architecture and training parameters of both the DiT and the VAE.

### B.2.1    DiT

Building upon the GPT-2 [3] architecture, we developed the core framework of DiT [59], incorporating rotary positional encoding to enhance the model's capacity for sequence modeling. In addition, we

introduced a timestep feature injection mechanism, which deeply integrates temporal information into the inference process, thereby aligning more effectively with the denoising dynamics of diffusion models.

To address the challenge of limited GPU memory during long audio generation, we adopted a chunk-wise cascaded inference strategy. Specifically, inspired by MuCodec's approach [40] to maintaining coherence, we use the tail features of the previous audio chunk as a prompt to guide the generation of the current chunk, ensuring smooth transitions between musical segments. Furthermore, we implemented a masking mechanism to distinguish between prompt regions and regions to be generated. By concatenating the prompt features, mask indicators, and the audio codec representations as the model's input, we significantly enhance the model's ability to generate long, coherent sequences.

Recognizing the distinct characteristics of mixed token and dual-track token audio representations, we designed differentiated network structures for each. In the mixed token mode, audio is first processed by a MuEncoder [40] and then discretized via RVQ to obtain a unified representation. In contrast, the dual-track mode involves separating vocals and accompaniment, each of which is independently encoded and discretized, resulting in two distinct representations. To accommodate these differing requirements, we tailored the model configurations accordingly: for the mixed token mode, we employed 1000 positional encodings, 39 transformer layers, 30 attention heads, and an embedding dimension of 1200; for the dual-track mode, we used 1000 positional encodings, 16 transformer layers, 20 attention heads, an embedding dimension of 2200, and set the inner layer dimension to 4400.

We trained our DiT model using a large-scale dataset of 300,000 music samples sourced from the internet, incorporating subsets of DISCO-10M [56], the Million Song Dataset [57], and our own in-house collections. From this dataset, we randomly selected 10,000 music samples to serve as our test set. For each training epoch, we randomly sampled 100,000 audio clips from the full training set, while 200 samples were drawn from the test set for evaluation during that epoch. For optimization, we employed the AdamW optimizer, setting the learning rate to $3 \times 10^{-5}$ and the weight decay to 1e-8. The optimizer's beta parameters were configured as $\beta_1 = 0.9$ and $\beta_2 = 0.999$, with an epsilon value of $10^{-8}$ to ensure numerical stability. Training was conducted with a batch size of 3 and gradient accumulation over 3 steps to effectively manage memory usage. For both the Music Codec (Mixed) and Music Codec (Dual-Track) models, we utilized 8 40GB NVIDIA A100 GPUs for each configuration, training each model's DiT for 300,000 steps.

### B.2.2 VAE

We retrained a 48kHz stereo Wave VAE based on Stable Audio [60], with a latent representation dimension of 64. Specifically, we set the strides of the encoder and decoder to 2, 4, 4, 6, and 10, resulting in a downsampling rate of 1920. Consequently, the original 48kHz audio is downsampled to a final frame rate of 25Hz, which is consistent with the frame rate of the muencoder. Apart from these adjustments, all other configurations—including model parameters for both the generator and discriminator, as well as training hyperparameters—remain identical to those used in Stable Audio[5].

The Wave VAE was trained for 1,000k steps on the same music dataset as LeLM on 8 40GB NVIDIA A100 GPUs. During DiT training, the parameters of the Wave VAE were kept frozen, and its latent representations of music were used as the training targets for DiT.

## C    Results of the Music Reconstruction

### C.1    Reconstruction quality

To comprehensively and thoroughly evaluate the performance of Music Codec, we selected several representative and high-performing audio codecs for comparison, including SemantiCodec [61], WavTokenizer [62], XCodec [63], and MuCodec [40]. Following the experimental setup of MuCodec, we focused on two key aspects: vocals (measured by speaker embedding similarity, SPK_SIM, and word error rate, WER) and accompaniment (measured by the VISQOL metric). This allows

---

[5]`https://github.com/Stability-AI/stable-audio-tools/blob/main/stable_audio_tools/configs/model_configs/autoencoders/stable_audio_2_0_vae.json`

for a more detailed assessment of Music Codec's performance across different dimensions. The corresponding experimental results are presented in Table 5.

Table 5: Comprehensive comparison of Codec reconstruction quality across bitrates.

| Method | CodeBook | Tokenrate (tps) | Bitrate (kbps) | VISQOL ↑ | SPK_SIM ↑ | WER (%) |
|---|---|---|---|---|---|---|
| Original music | — | — | — | — | — | 10.92 |
| SemantiCodec | 1 x 32768 | 25 | 0.375 | 1.92/1.92 | 0.52 | 120.17 |
|  | 1 x 16384 | 100 | 1.40 | 1.96/1.96 | 0.68 | 55.17 |
| WavTokenizer | 1 x 4096 | 40 | 0.48 | 2.93/2.93 | 0.49 | 101.49 |
|  | 1 x 4096 | 75 | 0.90 | 3.05/3.05 | 0.56 | 86.19 |
| XCodec | 1 x 1024 | 50 | 0.50 | 3.04/3.04 | 0.53 | 85.10 |
|  | 2 x 1024 | 50 | 1.00 | 3.30/3.30 | 0.79 | 55.37 |
|  | 4 x 1024 | 50 | 2.00 | 3.38/3.38 | 0.63 | 36.32 |
|  | 8 x 1024 | 50 | 4.00 | 3.58/3.58 | 0.72 | 26.42 |
| MuCodec | 1 x 16384 | 25 | 0.35 | 3.17/3.18 | 0.75 | 36.21 |
|  | 4 x 10000 | 25 | 1.33 | 3.45/3.46 | 0.87 | 24.26 |
| Music Codec (Mixed) | 1 x 16384 | 25 | 0.35 | 3.27/3.27 | 0.78 | 38.22 |
|  | 2 x 16384 | 25 | 0.70 | 3.34/3.34 | 0.82 | 33.43 |
|  | 4 x 16384 | 25 | 1.40 | 3.52/3.53 | 0.84 | 28.92 |
| Music Codec (Dual-Track) | 16384+16384 | 25 | 0.70 | 3.43/3.44 | 0.82 | 31.54 |

In low-bitrate scenarios (bitrate < 0.5 kbps), both MuCodec and Music Codec demonstrate clear advantages over WavTokenizer, SemantiCodec, and XCodec. Whether in terms of preserving vocal characteristics or reconstructing background quality, MuCodec and Music Codec consistently achieve superior scores. For example, at a bitrate of 0.35 kbps, both Music Codec (Mixed) and MuCodec significantly outperform other methods in terms of VISQOL, SPK_SIM, and WER, fully showcasing their reconstruction capabilities at extremely low bitrates.

As the bitrate increases, the reconstruction performance of XCodec, WavTokenizer, and SemantiCodec improves in both vocal and background dimensions. However, they still lag behind MuCodec and Music Codec at comparable bitrates. For instance, at around 0.9 kbps, WavTokenizer shows improvements in VISQOL and SPK_SIM, but its WER remains higher than that of MuCodec and Music Codec. Similarly, XCodec achieves a VISQOL of 3.58 at 4.0 kbps, yet its WER is still higher than that of Music Codec at much lower bitrates. These results indicate that MuCodec and Music Codec not only enhance audio quality and intelligibility but also maintain better bitrate efficiency.

A closer comparison between MuCodec and Music Codec (Mixed) reveals that both employ a hybrid representation to model vocals and background, and their overall performance is similar at the same bitrate (e.g., 0.35 kbps). However, Music Codec (Mixed) is slightly inferior to MuCodec in terms of the VISQOL metric. This can be attributed to MuCodec's use of the Mel-spectrogram as an intermediate representation. Compared to directly mapping latent features to waveforms, using the Mel-spectrogram as a bridge provides stronger expressive power for audio quality reconstruction.

Furthermore, it is noteworthy that Music Codec (Dual-Track) achieves even better performance than Music Codec (Mixed) at the same bitrate of 0.70 kbps across all metrics. Although both operate at the same bitrate, the dual-Track approach explicitly models vocals and background separately, which allows for more accurate reconstruction of audio details and structure compared to the hybrid approach. This is also one of the main reasons why we ultimately adopt dual-track tokens as the prediction target in the LeVo system.

## C.2 Reconstruction efficiency

In this experiment, we conducted a comparative analysis of the inference speed between MuCodec and Music Codec (Dual-Track), with detailed results presented in Table 6. We adopted the Real-

Time Factor (RTF) as the evaluation metric and divided the audio reconstruction process into three stages: first, Wav2Code, where the Codec Encoder transforms audio into discrete codes; second, Code2Latent, in which the DiT model performs multi-step denoising to convert the codes into latent representations; and finally, Latent2Wav, where the VAE decoder reconstructs the audio from the latent representations. To ensure a fair comparison, we set the number of inference steps for both models to 50.

Table 6: Comparison of Inference Efficiency between MuCodec and Music Codec (Dual-Track).

| Method | DiT Param | Real-Time Factor | | | Total |
| --- | --- | --- | --- | --- | --- |
| | | Wav2Code | Code2Latent | Latent2Wav | |
| MuCodec | 743M | 0.0201 | 0.5576 | 0.1001 | 0.6778 |
| Music Codec (Dual-Track) | 770M | 0.0183 | 0.0680 | 0.0039 | 0.0902 |

As shown in Table 6, MuCodec and Music Codec (Dual-Track) have very similar DiT parameter sizes, at 743M and 770M, respectively. In the Wav2Code stage, their RTFs are 0.0201 and 0.0183, indicating almost identical efficiency in the encoder component. However, in the Code2Latent stage, MuCodec's RTF reaches 0.5576, while Music Codec (Dual-Track) achieves a much lower 0.0680—an almost eightfold difference—demonstrating a significant advantage for Music Codec (Dual-Track) in generating latent representations through denoising. Furthermore, in the Latent2Wav stage, MuCodec's RTF is 0.1001, compared to just 0.0039 for Music Codec (Dual-Track), a similarly striking gap. This improvement can be largely attributed to the fact that Music Codec (Dual-Track) establishes a direct mapping from latent representations to audio, bypassing the intermediate mel-spectrogram representation and thus greatly reducing inference time.

Overall, Music Codec (Dual-Track) significantly outperforms MuCodec in terms of total inference speed, with a total RTF of only 0.0902 compared to MuCodec's 0.6778—a nearly sevenfold difference. This demonstrates that, given similar parameter sizes, Music Codec (Dual-Track) offers a clear advantage in inference efficiency. Moreover, as shown in the results of Table 1, its reconstruction performance is highly comparable to that of MuCodec. This combination of speed and quality is the primary reason we have chosen Music Codec (Dual-Track) as the core decoding module in the LeVo system.

## D   Detailed Experimental Settings

For evaluation, we generate 20 distinct lyrics and accompanying text descriptions with a large language model, and select 20 unseen music clips as audio prompts. For every lyric we produced up to three song variants, depending on system capabilities: (i) lyric-only, (ii) lyric + text description, and (iii) lyric + audio prompt. All metrics were computed over the entire set of generated songs.

### D.1   Details in objective evaluations

Here, we provide details of the objective evaluations.

**FAD**   Fréchet Audio Distance (FAD) is used to evaluate the generation fidelity of music. We calculate FAD based on the distribution distance between the feature of the generated audio and MusicCaps [14] test set using audioldm_eval[6].

**PER**   Phoneme Error Rate (PER) is used to evaluate the lyrics alignment and intelligibility of songs. To calculate PER, the vocal track is first extracted by Demucs, and then Whisperlarge-v2 [47] is utilized for lyric recognition. Using the clean vocal track rather than the song audio improves recognition accuracy. Whisperlarge-v2 is one of the state-of-the-art automatic speech recognition (ASR) models. It not only achieves high performance for speech but is also robust in recognizing singing voice. Considering singing often involves homophones or elongated vowels that do not alter

---

[6]https://github.com/haoheliu/audioldm_eval

phonetic content. Word-level (WER) or character-level (CER) metrics, therefore, penalize many errors that are inaudible to listeners. By operating at the phoneme level, PER captures intelligibility more faithfully and correlates better with human ratings of lyric clarity.

**MuQ-T and MuQ-A**   To quantify how faithfully LeVo follows user prompts, we report two embedding-based similarity scores derived from MuQ-MuLan [41], a contrastive music–language pre-training model.

- **MuQ-T**: similarity between the song and its text description.
- **MuQ-A**: similarity between the song and its audio prompt.

It is important to note that while the CLAP [64] is commonly used for similarity calculations, the training set of CLAP is dominated by non-vocal audio events, leading to inaccurate similarity comparisons. For a clean evaluation, MuQ-T is calculated on songs generated only from the text description, and MuQ-A on songs generated only from the audio prompt, avoiding conflicts between mismatched prompt modalities.

**Meta Audiobox-Aesthetic**   Measures the perceived musical aesthetics by leveraging advanced neural networks, including content enjoyment (CE), content usefulness (CU), production complexity (PC), and production quality (PQ).

### D.2   Details in subjective evaluations

Table 7: Subjective results of comparison and ablation systems for song generation on OVL, MEL, and HAM with 95% Confidence Intervals. The overall first and second results are marked with **bold** and underline, respectively.

| Models | MOS $\uparrow$ | | |
|---|---|---|---|
| | **OVL** | **MEL** | **HAM** |
| Suno-V4.5 | $\mathbf{3.59 \pm 0.054}$ | $\mathbf{4.10 \pm 0.042}$ | $\mathbf{3.93 \pm 0.033}$ |
| Haimian | $3.05 \pm 0.044$ | $3.51 \pm 0.046$ | $3.55 \pm 0.049$ |
| Mureka-O1 | $\underline{3.42 \pm 0.047}$ | $3.88 \pm 0.049$ | $3.89 \pm 0.030$ |
| YuE | $2.45 \pm 0.053$ | $3.04 \pm 0.052$ | $2.94 \pm 0.058$ |
| DiffRhythm | $2.60 \pm 0.054$ | $3.18 \pm 0.051$ | $3.22 \pm 0.066$ |
| ACE-Step | $2.26 \pm 0.065$ | $3.02 \pm 0.070$ | $3.30 \pm 0.073$ |
| SongGen* | $2.91 \pm 0.049$ | $3.43 \pm 0.053$ | $3.44 \pm 0.055$ |
| LeVo | $\underline{3.42 \pm 0.051}$ | $\underline{3.93 \pm 0.044}$ | $\underline{3.90 \pm 0.032}$ |
| w/o Train stage 2 | $3.29 \pm 0.044$ | $3.76 \pm 0.052$ | $3.77 \pm 0.044$ |
| w/o AR decoder | $2.93 \pm 0.053$ | $3.44 \pm 0.057$ | $3.34 \pm 0.073$ |
| w/o Dual-Track | $3.25 \pm 0.049$ | $3.82 \pm 0.048$ | $3.84 \pm 0.040$ |
| w/o DPO | $3.18 \pm 0.050$ | $3.71 \pm 0.061$ | $3.76 \pm 0.039$ |

For the subjective evaluations, we focus on the performance of generated songs in the following six dimensions:

- **Overall Quality (OVL).** The overall musicality and naturalness of the generated song.
- **Vocal Melodic Attractiveness (MEL).** The beauty, smoothness, and appeal of the vocal melody.
- **Vocal-Instrument Harmony (HAM).** The coherence and integration between vocals and accompaniment.
- **Song Structure Clarity (SSC).** The clarity and distinction of song sections (e.g., verse, chorus, bridge)
- **Audio Sound Quality (AQ).** The clarity, absence of noise, dynamic range, and technical quality of the audio.

Table 8: Subjective results of comparison and ablation systems for song generation on SSC, AQ, and LYC with 95% Confidence Intervals. The overall first and second results are marked with **bold** and underline, respectively.

| Models | MOS ↑ | | |
|---|---|---|---|
| | **SSC** | **AQ** | **LYC** |
| Suno-V4.5 | **4.19 ± 0.077** | **4.00 ± 0.068** | 3.17 ± 0.067 |
| Haimian | 3.62 ± 0.060 | 3.87 ± 0.040 | 3.32 ± 0.062 |
| Mureka-O1 | 4.14 ± 0.077 | 3.87 ± 0.068 | 3.32 ± 0.056 |
| YuE | 3.53 ± 0.062 | 3.08 ± 0.055 | 2.41 ± 0.060 |
| DiffRhythm | 3.55 ± 0.060 | 3.09 ± 0.054 | 2.69 ± 0.050 |
| ACE-Step | 3.21 ± 0.092 | 2.36 ± 0.064 | 2.22 ± 0.060 |
| SongGen* | 3.66 ± 0.061 | 3.69 ± 0.054 | 2.84 ± 0.063 |
| LeVo | 4.09 ± 0.060 | 3.96 ± 0.038 | **3.38 ± 0.071** |
|    w/o Train stage 2 | 3.80 ± 0.052 | 3.96 ± 0.037 | 2.91 ± 0.072 |
|    w/o AR decoder | 3.59 ± 0.044 | 3.71 ± 0.051 | 2.74 ± 0.058 |
|    w/o Dual-Track | 3.96 ± 0.058 | 3.86 ± 0.038 | 3.18 ± 0.073 |
|    w/o DPO | 3.97 ± 0.068 | 3.93 ± 0.041 | 3.18 ± 0.078 |

- **Lyrics Following Accuracy (LYC).** The accuracy and clarity with which the vocals follow the intended lyrics.

We conducted MOS (Mean Opinion Score) tests for all aspects, providing subjects with detailed descriptions, and reported both mean and CI95 scores of our MOS tests. The subjects present and rate the samples, and each subject is asked to evaluate on a scale of 1-5. Our MOS tests were conducted by 10 music professionals. All the screenshots of instructions for subjects have been shown in Figure 4-9. We paid at least the minimum wage in the country of these subjects. We tell the subjects that the data will be used in scientific research.

# E  Detailed Results with 95 % Confidence Intervals

To support the main claims of the paper, we report in Tables 7 and 8 the complete MOS scores together with their 95 % confidence intervals (CI95). Table 7 presents MOS for Overall quality (OVL), Vocal Melodic Attractiveness (MEL), and Vocal-Instrument Harmony (HAM). Table 8 lists MOS for Song Structure Clarity (SSC), Audio Sound Quality (AQ), and Lyrics Alignment Accuracy (LYC). For objective evaluations, the error bars are very small due to the large number of test samples. Specifically, at a 95% confidence level, the error margin for MuQ-T and MuQ-A scores across all models is less than 0.01, and for content enjoyment (CE), content usefulness (CU), and production quality (PQ), the error margin is less than 0.015. For production complexity (PC), the error margin is less than 0.05. The narrow intervals confirm that inter-rater variability is low and that the trends reported in Section 4 are statistically reliable:

- **Superiority of LeVo.** LeVo's MOS scores remain significantly higher than every open-source system across all six dimensions. The non-overlapping CIs corroborate LeVo's superiority.
- **Competitiveness with industry systems.** While Suno-V4.5 leads on most dimensions, its CIs overlap with (or are exceeded by) LeVo's on many dimensions, underscoring our claim of state-of-the-art alignment performance even against proprietary systems.
- **Effectiveness of each module.** Ablation rows show consistent, statistically meaningful drops. These non-overlapping intervals validate the contribution of every design choice.

# F  Contribution of Each Module to Audio Quality

To investigate how different components of LeVo contribute to the final audio quality, we conducted a Mean Opinion Score (MOS) listening test on 40 unseen songs. The evaluated audio included:

Table 9: Subjective results on audio quality generated by different modules within LeVo.

| Models | AQ(Audio Quality) |
|---|---|
| GT | 3.81 |
| VAE Recon | 3.79 |
| Codec Recon | 3.75 |
| LeLM | 0.21 |

(1) ground-truth recordings, (2) VAE-reconstructed audio, (3) Codec-reconstructed audio, and (4) LeVo-generated audio using the same lyrics. As shown in Table 9, the overall degradation in audio quality mainly arises from three sources: VAE reconstruction loss (0.02), Codec reconstruction loss (0.04), and LeLM modeling loss (0.09). Among these, the loss introduced by LeLM modeling is the most pronounced. While LeVo achieves higher perceived audio quality than models such as SongGen and LeVo w/o AR Decoder—both of which generate dual-track tokens—it still lags behind the Codec reconstruction, revealing room for improvement in the generative modeling stage. In addition, both the VAE and Codec modules contribute to minor degradation during the token-to-audio process.

# G   Smooth Transitions in Interpolation-based Multi-Preference Alignment Method

To further demonstrate the flexibility of our proposed interpolation-based multi-preference alignment method, we measured its behavior under four weight settings that progressively move from no preference control to a uniform blend of the three preferences. When the model is trained without DPO, it serves as the baseline for all subsequent comparisons. The results are presented in Table 10. Steering the model with a single preference—that is, assigning an interpolation coefficient of (1, 0, 0) for Lyric Alignment Preferences (Strategy 1), (0, 1, 0) for Prompt Consistency Preference (Strategy 2), or (0, 0, 1) for Musicality Preference (Strategy 3) enhances exactly the targeted dimension. Concretely, Strategy 1 reduces the phoneme error rate (PER) from 10.6% to 7.0%, Strategy 2 raises MuQ-T and MuQ-A to 0.34 and 0.83, respectively, and Strategy 3 delivers the highest Audiobox-Aesthetic scores. When two preferences are mixed with equal weights, the model exhibits smooth trade-offs: combining Strategy 1 and Strategy 2 slightly relaxes PER (from 6.5 % to 7.0 %) compared to using Strategy 1 alone, but yields noticeable gains in MuQ-T and MuQ-A; blending Strategy 1 with Strategy 3 preserves most of the PER benefit while further improving Audiobox-Aesthetic scores compared to using Strategy 1 alone; pairing Strategy 2 with Strategy 3 simultaneously maintains high MuQ scores and Audiobox-Aesthetic scores. Finally, the uniform three-way interpolation (0.33, 0.33, 0.33) produces consistent improvements across all metrics. Taken together, these results verify that by simply adjusting the inference-time interpolation coefficients (without retraining or additional fine-tuning), this approach can achieve a smooth transition in performance to satisfy diverse application needs more flexibly than conventional data-mixing approaches.

Table 10: Comparison of various preference combinations across multiple objective metrics. The overall first and second results are marked with **bold** and underline, respectively.

| Models | FAD ↓ | MuQ-T ↑ | MuQ-A ↑ | PER ↓ | Content Scores ↑ | | | |
|---|---|---|---|---|---|---|---|---|
| | | | | | CE | CU | PC | PQs |
| w/o DPO | **2.60** | 0.31 | 0.82 | 10.6 | 7.70 | 7.86 | 5.89 | 8.39 |
| Strategy 1 | 2.85 | 0.30 | 0.81 | **6.5** | 7.72 | 7.86 | 5.97 | 8.42 |
| Strategy 2 | 2.89 | **0.34** | **0.83** | 10.3 | 7.75 | 7.87 | 5.96 | 8.43 |
| Strategy 3 | 2.63 | 0.32 | 0.82 | 11.2 | **7.78** | **7.93** | **6.16** | 8.45 |
| Strategy 1&2 | 2.91 | 0.33 | **0.83** | 7.0 | 7.67 | 7.83 | 5.93 | 8.40 |
| Strategy 1&3 | 2.75 | 0.31 | 0.82 | 8.3 | 7.76 | 7.90 | 6.10 | 8.44 |
| Strategy 2&3 | 2.73 | **0.34** | **0.83** | 10.9 | 7.75 | 7.92 | 6.01 | 8.43 |
| Strategy 1&2&3 | 2.68 | **0.34** | **0.83** | 7.2 | **7.78** | 7.90 | 6.03 | **8.46** |

Table 11: Pearson Correlation between subjective evaluation results and Meta Audiobox-Aesthetic metrics.

|    | OVL  | MEL  | HAM   | SSC  | AQ    | LYC   |
|----|------|------|-------|------|-------|-------|
| CE | 0.32 | 0.44 | 0.28  | 0.07 | 0.31  | −0.01 |
| CU | 0.26 | 0.36 | 0.16  | 0.02 | 0.27  | −0.14 |
| PC | −0.08 | −0.14 | 0.32 | 0.44 | −0.14 | 0.23  |
| PQ | 0.21 | 0.07 | −0.06 | 0.01 | 0.15  | −0.12 |

## H    Risks of Memorization and Content Leakage

We further assess the potential risks of memorization and content leakage in LeVo. To examine whether the model inadvertently reproduces training data, we randomly selected 40 songs from the training set and generated corresponding outputs using the same input conditions. We then evaluated the similarities between the generated and original songs by computing the 5-gram overlap and Levenshtein distance similarity of the extracted dual-track token sequences, supplemented by manual inspection of melodic patterns. The quantitative results show extremely low similarity scores—0.0001 for 5-gram and 0.0012 for Levenshtein distance—and no perceptually similar melodies were identified through human evaluation. These results suggest that LeVo does not memorize or directly replicate its training data, even when prompted with identical inputs.

We attribute this low memorization tendency to two main factors. First, the relatively modest model scale (2B parameters) compared to the large training corpus of 2 million songs reduces the likelihood of overfitting to specific samples. Second, the multi-preference alignment stage following pre-training serves as an additional regularization process, further mitigating memorization by optimizing the model under diverse alignment objectives rather than maximizing likelihood on training data. Together, these results indicate that LeVo maintains a low risk of content leakage, supporting its reliability for safe and responsible music generation.

## I    Correlation between Human Evaluation Results and Objective Metrics

To better analyze the limitations of current objective metrics in music generation, we compared subjective and objective evaluation results. Overall, we observed a noticeable discrepancy between human judgments and automatic metrics. To quantify this relationship, we calculated the Pearson Correlation Coefficients between human evaluation results and objective scores.

A weak correlation was found between FAD and AQ ($r = −0.24$), suggesting that distribution-level metrics such as FAD may fail to capture subtle acoustic differences that influence human perception. In contrast, PER and LYC exhibit a strong correlation ($r = −0.77$), indicating that ASR-based metrics effectively reflect lyric-following accuracy. Nevertheless, these metrics are still affected by factors such as background accompaniment and vocal pitch, which can introduce recognition errors.

For the Meta Audiobox-Aesthetic metrics and subjective evaluation results, the correlations were shown in Table 11. The results show that content enjoyment (CE) exhibits the strongest correlations with OVL (0.32), MEL (0.44), and AQ (0.31), suggesting that these objective metrics are most aligned with human enjoyment of music. Content usefulness (CU) shows similar but slightly weaker patterns. Production complexity (PC) correlates most strongly with HAM (0.32) and SSC (0.44), indicating sensitivity to variations in vocal and accompaniment elements. In contrast, production quality (PQ) shows only weak correlations with all subjective dimensions, reflecting its limited ability to capture human-perceived quality.

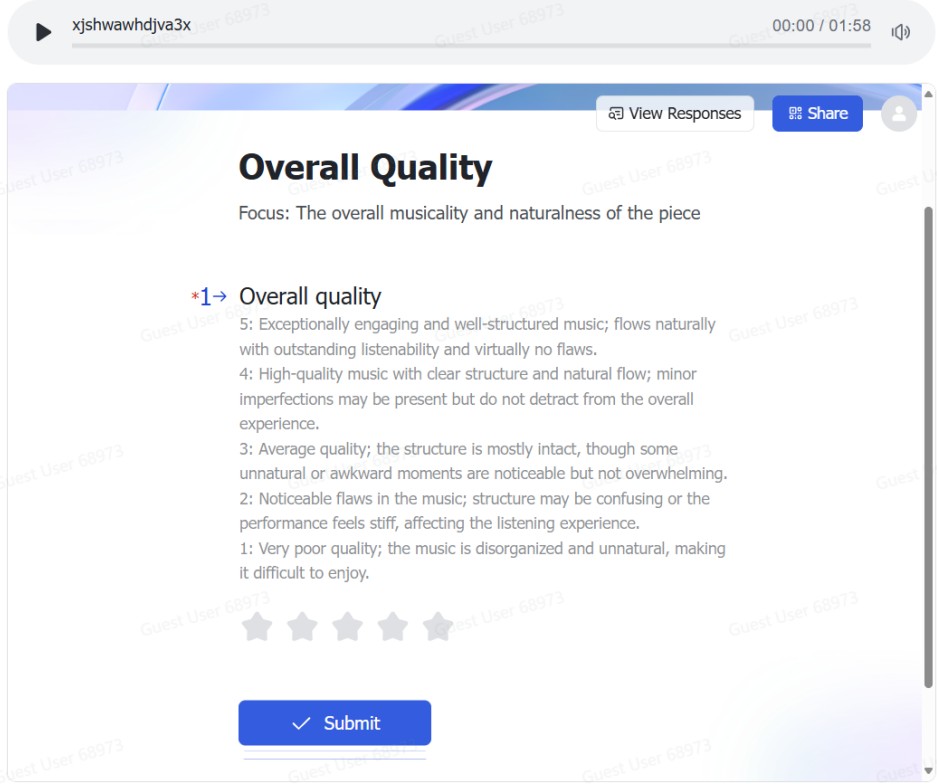

Figure 4: The screenshot of MOS test in overall quality.

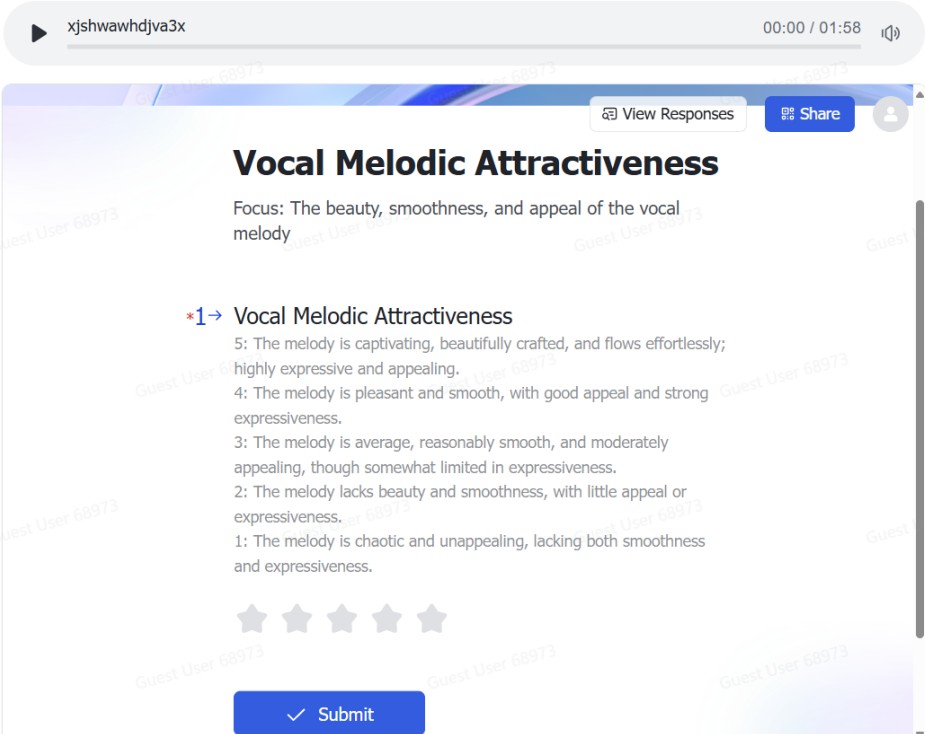

Figure 5: The screenshot of MOS test in vocal melodic attractiveness.

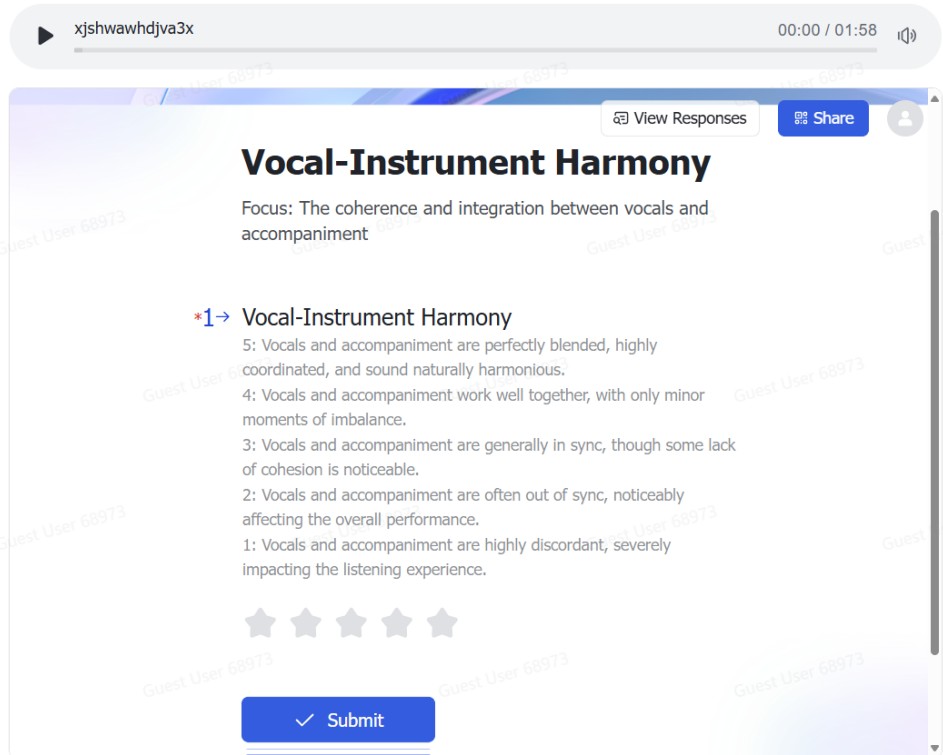

Figure 6: The screenshot of the MOS test in vocal-instrument harmony.

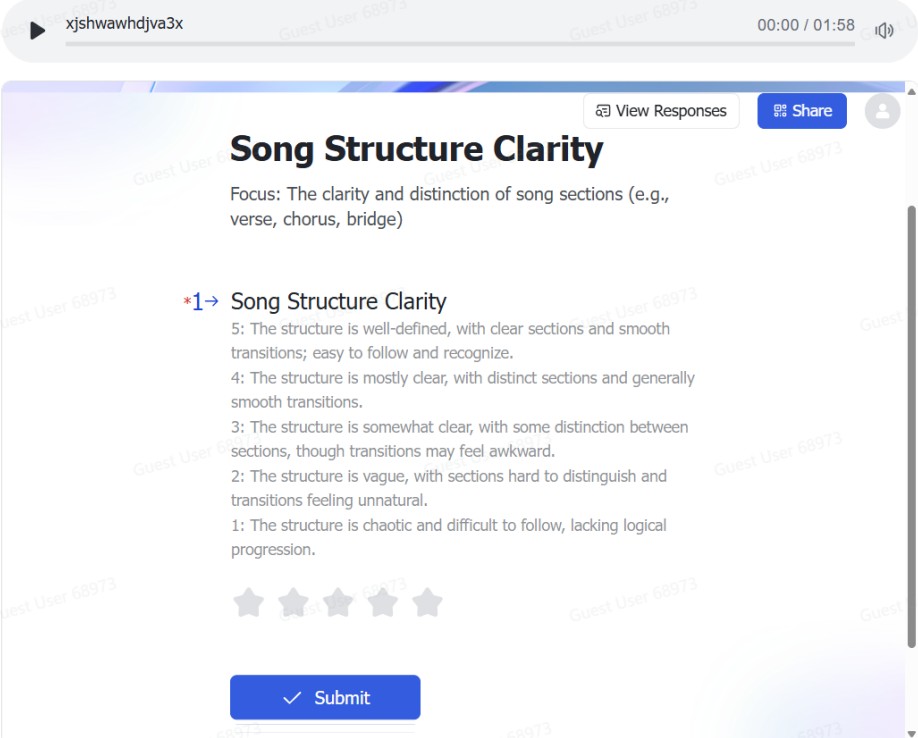

Figure 7: The screenshot of the MOS test in song structure clarity.

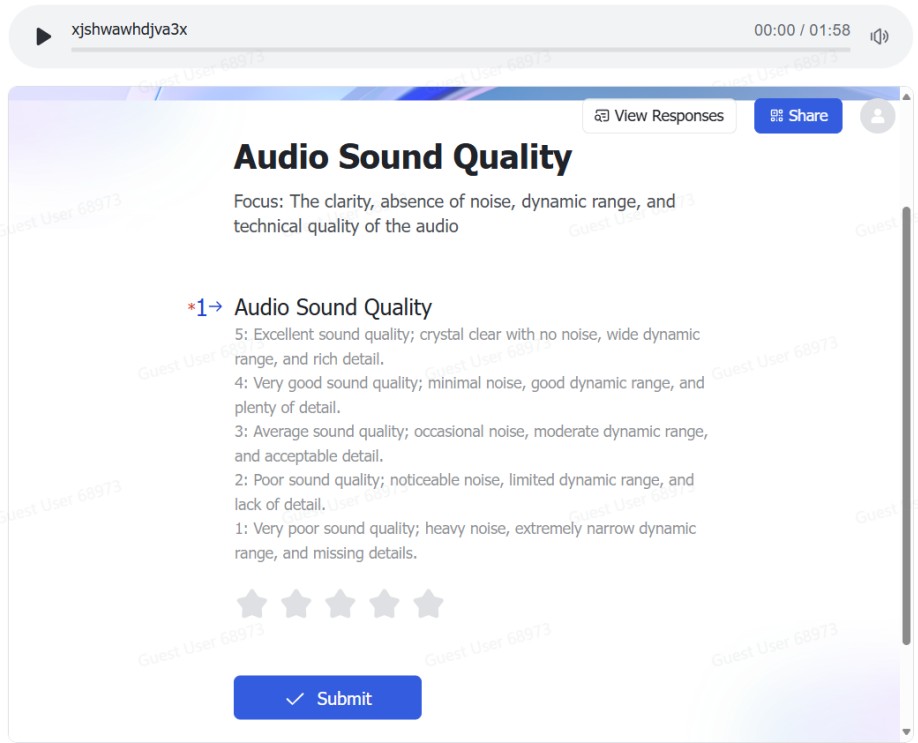

Figure 8: The screenshot of the MOS test in audio sound quality.

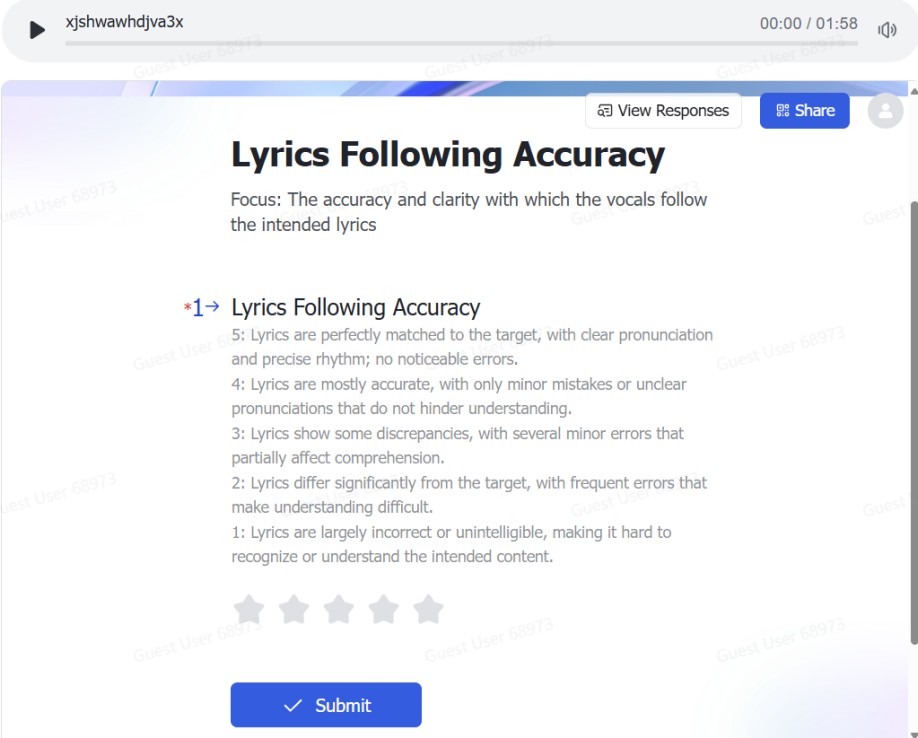

Figure 9: The screenshot of the MOS test in lyrics following accuracy.

