# OpenReview forum: "LeVo: High-Quality Song Generation with Multi-Preference Alignment"
_NeurIPS.cc/2025/Conference — NeurIPS 2025 poster_

### Official Review · Reviewer_7iGT · 2025-07-02

**Clarity:** 4
**Significance:** 3
**Originality:** 3
**Rating:** 5
**Confidence:** 5

**Summary:**

This paper presents LeVo, a conditional music generation model. The authors propose a three-stage training pipeline: (1) autoregressive pretraining on mixed audio tokens, (2) fine-tuning on separate vocal and accompaniment tokens to improve fidelity and musicality, and (3) preference alignment using direct preference optimization (DPO) to enhance lyrics alignment, prompt consistency, and musicality. LeVo addresses key limitations of prior works, including low audio fidelity, vocal-instrument mismatch, and noisy training data. Extensive experiments and ablation studies validate the effectiveness of the proposed methods.

**Questions:**

- How are the residual vector quantization (RVQ) tokens handled during training and inference? Are they flattened into a sequential token stream or processed in parallel (e.g., stacked across channels)?
- What types of prompts were used for evaluation? Were they sufficiently diverse in terms of genre, emotion, and complexity to rigorously assess the model’s capabilities across different generation scenarios?
- Is the overall audio quality constrained by the capacity of the Stable Audio VAE decoder used in the codec? Or is there evidence that further improvements can be achieved independent of the codec's ceiling? A more detailed analysis of each module’s contribution to the final audio quality would be valuable for understanding where the main bottlenecks lie.
- Line 312: AR decoder?

**Ethical Concerns:**

["Major Concern: Data privacy, copyright, and consent"]

**Final Justification:**

The paper is clearly written and well-structured. The motivation and proposed methods are logically presented and easy to follow. The literature review is thorough and well-contextualized within the scope of related work. The proposed approach, which leverages dual-track tokens and Direct Preference Optimization (DPO), effectively addresses the challenges of text-to-music generation.
The authors have addressed all of the questions I previously raised. Additionally, they conducted a correlation analysis between subjective and objective metrics and evaluated potential memorization of training data to further clarify and support their results.

**Limitations:**

yes

**Paper Formatting Concerns:**

.

**Quality:**

3

**Strengths And Weaknesses:**

Strengths
- The paper is clearly written and well-structured. The motivation and proposed methods are easy to follow and logically presented.
- The literature review is thorough and well-contextualized.
- The authors conduct comprehensive objective and subjective evaluations, including careful ablation studies, to support the effectiveness of their approach.
- The paper makes effective use of a variety of pretrained models (e.g., DEMUCS, MuQ-MuLan, MuCodec, All-in-One, Qwen2-Audio, Whisper) for both data preparation and evaluation. These choices are well-justified and help address the limitations of noisy or incomplete training data.

Weaknesses
- The reliance on multiple pretrained models introduces a dependency on their quality. For instance, LeVo’s relatively lower performance in song structure clarity may be due to limitations in the All-in-One model. A deeper analysis of how such dependencies impact final performance would strengthen the work.
- There are notable discrepancies between objective and subjective evaluation results (e.g., PER vs. LYC), but the paper lacks sufficient discussion to interpret these gaps. A more nuanced analysis would both guide future research and clarify the limitations of current evaluation metrics.
- The MOS evaluation lacks clear calibration or reference targets. Including real human or professionally produced examples in the listening test could help contextualize the reported scores.
- The training data includes licensed materials (e.g., DISCO-10M, Million Song Dataset), but the paper does not discuss the ethical implications or risk of memorization. An analysis of potential overfitting or content leakage would improve the ethical transparency of the work.

---

> ### Author Rebuttal · Authors · 2025-07-31
>
> We sincerely appreciate your time and efforts in reviewing our paper. Your constructive feedback has significantly improved our work. Below, we provide detailed responses to your concerns and questions.
>
> **Regarding Weakness 1: Dependency on** **Pre-trained Model** **Quality**
>
> We sincerely appreciate your valuable suggestion. Due to the lack of labeled music data and the high cost of annotation, we have relied on pre-trained models for data labeling. As you mentioned, the quality of these models can affect the overall performance. Our experiments on a small dataset revealed that the current pre-trained models achieve around 76% accuracy in song structure labeling and approximately 50% accuracy in song description labeling. In contrast, LeVo achieves 80% accuracy in song structure and 52% in song description. This initial experiment shows that, with our proposed model structure, training methods, and DPO-based multi-preference alignment method, LeVo's song structure clarity and prompt adherence surpass the accuracy of the labeling models. However, due to the limitations of pre-trained models, there is still a gap between these aspects and current state-of-the-art industry systems. We will further investigate this dependency and add more accurate and extensive experimental results in the final version of the paper. Additionally, we are currently working to improve the labeling accuracy of pre-trained models and introducing human-annotated data for fine-tuning to explore how to enhance model performance further.
>
> **Regarding Weakness 2: Discrepancy between Objective and Subjective Evaluation Results**
>
> As the reviewer mentioned, objective metrics in music evaluation have limitations, which result to the observed differences between objective and subjective results. Given that music is an auditory art, we set up multiple dimensions for subjective experiments to provide a more accurate and persuasive human auditory perception. To explore the limitations of objective metrics, we calculated the Pearson Correlation between human evaluation results and objective metrics. We found a weak correlation between FAD and AQ (-0.24), suggesting that distribution-level metrics may not capture subtle acoustic differences. However, there is a strong correlation between PER and LYC (-0.77), indicating that ASR models can capture the lyrics following accuracy. Nevertheless, ASR models are affected by background music, vocal pitch, and other factors, leading to some errors.
>
> For the Meta Audiobox-Aesthetic scores and the subjective evaluation results, the following correlations were observed:
>
> |      | OVL   | MEL   | HAM   | SSC  | AQ    | LYC   |
> | ---- | ----- | ----- | ----- | ---- | ----- | ----- |
> | CE   | 0.32  | 0.44  | 0.28  | 0.07 | 0.31  | -0.01 |
> | CU   | 0.26  | 0.36  | 0.16  | 0.02 | 0.27  | -0.14 |
> | PC   | -0.08 | -0.14 | 0.32  | 0.44 | -0.14 | 0.23  |
> | PQ   | 0.21  | 0.07  | -0.06 | 0.01 | 0.15  | -0.12 |
>
> The results indicate that CE (content enjoyment) has the strongest correlations with OVL (0.32), MEL (0.44), and AQ (0.31), indicating that CE is likely linked to human preferences for music. CU (content usefulness) also shows some correlations in these aspects but weaker than CE. PC (production complexity) has the strongest correlations with HAM (0.32) and SSC (0.44), showing sensitivity to variationgs in vocal and accompaniment elements. PQ (production quality) shows weak correlations with most subjective results, indicating limitations in capturing these six subjective properties. We will include this analysis in the final version of the paper to clarify the limitations of current metrics.
>
> **Regarding Weakness 3: MOS Evaluation Lacking Clear Calibration or Reference Targets**
>
> Thanks for your constructive suggestion. As commercial products like Suno and Haimian often prohibit the input of real song lyrics to prevent content leakage, we followed the experimental setups from MusiCot [17] and YUE [18], using lyrics generated by LLMs for the evaluation. During the experiments, we asked participants to listen to the entire music generated by different models with the same input before scoring, thereby obtaining scores that reflect the relative differences between models. Your suggestion is valuable, and in future work, we will attempt to manually produce a set of songs based on the generated lyrics for more accurate MOS evaluation.
>
> **Regarding Weakness 4 and Ethical Concerns: Ethical Implications and Risk of Memorization**
>
> As the reviewer rightly pointed out, the risks of memorization and content leakage are important issues in song generation, and they have often been overlooked in prior work. We conducted a new experiment where we selected 40 songs from the training set and generated corresponding songs using LeVo. We checked the similarities between the generated songs and the original ones by calculating 5-gram and Levenshtein distance similarity for the extracted dual-track token sequences and manually checking the similarity of the melodies.  The results showed that the 5-gram similarity was 0.0001, and the Levenshtein distance similarity was 0.0012, with no similar melodies found through manual inspection. These results indicate that using the same input is unlikely to cause training data leakage. This might be due to the relatively small model size (2B parameters) compared to the training dataset of 2 million songs, which prevents severe overfitting. Additionally, LeVo's multi-preference alignment stage after pre-training helps reduce the memorization risk. We will include these experimental results in the final version of the paper, and we hope to explore content leakage and better training strategies in future work.
>
> **Regarding Question 1: Handling of RVQ Tokens**
>
> For the mixed token, its primary role is to guide the overall arrangement of melody, rhythm, and tempo, so we only predict the first layer of RVQ tokens. For dual-track tokens, as shown in Table 2 of Appendix B, we found that using a 2-layer dual-track token (one for vocals and one for accompaniment) closely matches the performance of a 4-layer mixed token. Therefore, in our experiments, we used a 2-layer dual-track token, and these layers were processed in parallel through cross-channel stacking.
>
> **Regarding Question 2: Evaluation of Prompts**
>
> For audio prompts, we randomly selected unseen music segments from different genres for evaluation. For text descriptions, we used a LLM to generate diverse content, covering a wide range of genres, emotions, and complexities to rigorously assess the model's capabilities across different scenarios.
>
> **Regarding Question 3: Contribution of Each Module to Audio Quality**
>
> Thanks for your insightful suggestion. We conducted a new experiment where we selected 40 unseen songs and performed a MOS listening test on the audio sound quality of ground truth audio, VAE-reconstructed audio, Codec-reconstructed audio, and LeVo-generated audio (using the same lyrics). The results are as follows:
>
> | Model                | AQ   |
> | -------------------- | ---- |
> | Ground Truth         | 3.81 |
> | VAE Reconstruction   | 3.79 |
> | Codec Reconstruction | 3.75 |
> | LeVo                 | 3.66 |
>
> The results show that the overall loss in audio quality is primarily caused by three factors: VAE reconstruction loss (a decrease of 0.02), Codec reconstruction loss (a decrease of 0.04), and LeLM modeling loss (a decrease of 0.09). The loss caused by LeLM modeling is the most significant. While LeVo performs better than models like SongGen and LeVo w/o AR Decoder (both of which predict dual-track tokens), there is still a gap compared to Codec reconstruction. Additionally, the VAE module and Codec module also cause some loss when rendering tokens into audio. We will include this analysis in the final version of the paper and explore further improvements in audio quality from these two aspects in future work.
>
> **Regarding Question 4: Line 312 -** **AR** **Decoder**
>
> Thank you for pointing out the typo. We will correct this in the final version of the paper.
>
> **References:**
>
> [17] Lam, M. W. Y., Xing, Y., You, W., et al. "Analyzable chain-of-musical-thought prompting for high-fidelity music generation." *arXiv* *preprint* *arXiv:2503.19611*, 2025.
>
> [18] Yuan, R., Lin, H., Guo, S., et al. "YuE: Scaling open foundation models for long-form music generation." *arXiv* *preprint* *arXiv:2503.08638*, 2025.

---

> > ### Comment · Reviewer_7iGT · 2025-08-06
> >
> > Dear authors,
> >
> > Thank you for your clarification on my questions and for the deeper analysis of the evaluation.
> > With the updated analysis, the paper is now clearer and conveys reproducible insights to readers. I appreciate your contribution.

---

### Official Review · Reviewer_XdTb · 2025-07-03

**Clarity:** 3
**Significance:** 2
**Originality:** 3
**Rating:** 5
**Confidence:** 3

**Summary:**

This paper proposes a language model LeLM and a music codec for lyrics-to-song generation. The authors identify challenges in AI music generation such as synchronization loss, limited high quality data, inefficient prediction for long songs, and noisy or unreliable annotations. LeVo addresses these challenges by parallel prediction of mixed and dual-track tokens to guide overall structure while improving details. The authors also introduce a DPO-based multi-preference alignment method to fine tune for human preferences. Experiments show that LeVo has higher instruction-following ability, and achieves the highest Audiobox-Aesthetic scores among open-source approaches, and shows overall competitive results with academic baselines and industry models.

**Questions:**

What music data is the LeLM component trained on? Is the training data limited to Chinese and English songs?

I was not able to access the appendix. When I click on the supplementary materials link, it shows "no healthy upstream."

**Ethical Concerns:**

["NO or VERY MINOR ethics concerns only"]

**Final Justification:**

Thank you for the detailed rebuttal. Your explanation of how LeVo compares with other open-source approaches and industry systems addresses my concerns about its competitiveness despite the data limitations. I was able to access the appendix successfully this time, and I appreciate the additional detail provided there.

**Limitations:**

As the authors discussed, generated audio quality is constrained by data quality and leaves a performance gap compared to SOTA closed-source models. Further fine-tuning  is needed.

**Paper Formatting Concerns:**

None. The paper is well-written and clear, with a nicely formatted demo page.

**Quality:**

3

**Strengths And Weaknesses:**

Strengths:

(1) The idea of parallel mixed and dual-track token prediction with modular training is novel and addresses limitations of prior works like Jukebox and SongGen.

(2) The proposed multi-dimensional DPO fine-tuning is a novel and interesting extension of RLHF methods for AI music generation.

(3) Experimental results are strong, covering both objective and subjective evaluations, along with ablation studies.

Weaknesses:

(1) Performance of the proposed framework falls short of SOTA closed-source models like Suno.

(2) It is not clear whether the training data for LeLM covers languages beyond Chinese and English. If the dataset is bilingual, the model may have limited ability to generalize to other languages and cultural song styles.

---

> ### Author Rebuttal · Authors · 2025-07-31
>
> We are grateful for the reviewer’s overall positive response and the constructive comments provided. Below, we provide detailed responses to all the concerns you raised.
>
> **Regarding Weaknesses 1 and Limitations: Performance of the Proposed Framework Falls Short of** **SOTA** **Closed-Source Models like Suno**
>
> Due to copyright issues with music data, our model, as well as other studies such as MusiCot [17], YUE [18], and DiffRhythm [33], have relied primarily on open-source data and a small portion of in-house copyrighted data for training. These datasets are not only limited in quantity but also have lower quality (mainly sourced from non-professional singers), which significantly affects the musicality, background noise, and vocal pronunciation of the generated songs. In contrast, industry systems like Suno are likely able to access more extensive and higher-quality datasets.
>
> Although our performance is constrained by the quality of the data, our proposed model structure, training strategy, and the multi-preference alignment method allow LeVo to outperform existing open-source models in various dimensions. Moreover, LeVo is competitive with current state-of-the-art industry systems such as Mureka and Haimian, with close or superior performance to Suno in terms of vocal-instrument harmony, audio sound quality, and lyrics following accuracy. This demonstrates the potential of our approach. We believe that with access to licensed higher-quality data, LeVo could further improve. We will also explore increasing the model size, dataset size, and data quality in future work.
>
> **Regarding Weakness 2 and Question 1: The Component of** **Training Data**
>
> We sincerely appreciate your valuable suggestions regarding the training data. As mentioned in Appendix A (lines 6-7), our training data consists of licensed open-source datasets, such as DISCO-10M [1*] and the Million Song Dataset [2*], along with a small portion of copyrighted in-house data. Due to copyright limitations, the available data currently covers primarily Chinese and English songs, so LeVo is currently limited to these two languages.
>
> However, we believe that the structure and training methodology of LeLM are not language-dependent. Once enough data for other languages is available, we expect that the success seen in Chinese and English can be replicated in other languages. This limitation is solely due to the current availability of training data, and we plan to continuously collect data in more languages and demonstrate its capabilities across multiple languages in future work.
>
> **Regarding Question 2: Not Able to Access the Appendix**
>
> Thank you for your feedback. We have re-tested the link to the supplementary materials, and it successfully downloads and unzips the appendix. We suspect that the issue may have been caused by a network problem on your end. Since new links cannot be provided during the rebuttal phase, we kindly suggest that you attempts to click on the “Supplementary Material” link again to access it.
>
> **References:**
>
> [17] Lam, M. W. Y., Xing, Y., You, W., et al. "Analyzable chain-of-musical-thought prompting for high-fidelity music generation." *arXiv* *preprint* *arXiv:2503.19611*, 2025.
>
> [18] Yuan, R., Lin, H., Guo, S., et al. "YuE: Scaling open foundation models for long-form music generation." *arXiv* *preprint* *arXiv:2503.08638*, 2025.
>
> [33] Ning Z, Chen H, Jiang Y, et al. DiffRhythm: Blazingly fast and embarrassingly simple end-to-end full-length song generation with latent diffusion[J]. arXiv preprint arXiv:2503.01183, 2025.
>
> [1*] Lanzendörfer, L., Grötschla, F., Funke, E., et al. "Disco-10M: A large-scale music dataset." *Advances in Neural Information Processing Systems*, 2023, 36: 54451-54471.
>
> [2*] Bertin-Mahieux, T., Ellis, D. P. W., Whitman, B., & Lamere, P. "The Million Song Dataset." *In ISMIR*, volume 2, page 10, 2011.
>
> Note: [N] denotes the reference in the paper and [N*] represents the reference in the Appendix.

---

### Official Review · Reviewer_arNL · 2025-07-03

**Clarity:** 3
**Significance:** 3
**Originality:** 3
**Rating:** 5
**Confidence:** 4

**Summary:**

This paper explores direct preference optimization for lyric-conditioned music generation. The authors proposed a novel framework to construct training pairs for lyric alignment, prompt consistency and musicality preference modeling. The experimental results demonstrate the effectiveness of the proposed framework in improving the quality of the generated music with DPO, especially in terms of  lyric alignment. The ablation study also highlights the necessities of each proposed component.

**Questions:**

- (Line 71-73) "Compared to single token prediction (only using mixed or dual-track tokens) or straightforward joint training, our method simultaneously optimizes musicality, vocal–instrument harmony, and sound quality." -> Is this true? I thin the same applies to the baseline models.
- (Figure 2) The illustration for the prefix tokens is confusing. If I understand it correctly, they are fed as a sequence of tokens. Yet, the current illustration makes it look like they are being "summed" in some embedding space and fed to the model as one single token. It's also unclear to me what "Prompt Embedding" means here. Please clarify this.

**Ethical Concerns:**

["NO or VERY MINOR ethics concerns only"]

**Final Justification:**

The authors have addressed the issues of missing error bars and unsupported significance claims in their rebuttal. I think this paper introduces many new and reusable ideas to lyric-conditioned music generation, and am thus recommending accepting this paper.

**Limitations:**

Yes

**Quality:**

3

**Strengths And Weaknesses:**

### Strengths

- This paper clearly demonstrates the benefits of DPO for lyric-conditioned  music generation. The idea of adapting DPO for music generation is well executed, and the experimental results do show its effectiveness, most significantly observed in the improvement of lyric alignment.
- The authors compared the proposed method with many baseline models, including both industrial and academic models. It's impressive that the model can achieve competitive results to some of the leading commercial models in the subjective listening test.

### Weaknesses

- Error bars are missing, making it hard to see if any of these differences are significant.
- Table 1 is somewhat misleading without a row for "LeVo w/o DPO", i.e., the first row of Table 3. The effectiveness of the proposed DPO approach in improving PER is clear, yet its effectiveness in improving MuQ-T, MuQ-A and Meta's AudioBox-Aesthetic metrics are not quite clear as the numbers are close.
- Significance is claimed in multiple places without any significance test, including line 80 ("the proposed LeVo model significantly improves over the open-source music generation models ...") line 315 ("causes a significant decline in FAD, PER and PC ..."), and line 321 ("leads to significant decreases in OVL, MEL and LYC ..."). Please weaken these claim.
- Reproducibility can be a big issue. The dataset is massive with 2 million songs totaling 110k hours. Even if the code is released, there will be no easy way to reproduce the results from scratch.

---

> ### Author Rebuttal · Authors · 2025-07-31
>
> We thank the reviewer for recognizing our work. We appreciate the constructive comments provided, which will help us further improve our paper.  We hope our response below resolves your concerns and questions.
>
> **Regarding Weakness 1 and 3: Missing Significance Tests**
>
> We sincerely appreciate your valuable suggestions. To support the statistical significance of the differences in our subjective evaluations, we have added the complete MOS scores along with their 95% confidence intervals (CI95), as shown in Tables 4 and 5 in the Appendix. For objective evaluations, the error bars are very small due to the large number of test samples. Following prior works, such as MusiCot [17] and YUE [18], we did not include the error bars in the paper. Specifically, at a 95% confidence level, the error margin for MuQ-T and MuQ-A scores across all models is less than 0.01, and for content enjoyment (CE), content usefulness (CU), and production quality (PQ), the error margin is less than 0.015. For production complexity (PC), the error margin is less than 0.05. We thank you again for your suggestion, and we will include these results in the final version of the paper to further support our claims.
>
> **Regarding Weakness 2: Table 1 Misleading without "LeVo w/o DPO" Row**
>
> Thank you for your comment. In this paper, we placed the ablation results regarding DPO in Table 3. The first row of Table 3 corresponds to "LeVo w/o DPO", and the last row, "Interpolation", represents the LeVo model. We apologize for any confusion caused. In the final version, we will clearly label "LeVo (Interpolation)" and add a "LeVo w/o DPO" row to Table 1.
>
> Regarding the effectiveness of DPO in musicality, while the differences are not highly pronounced in the AudioBox-Aesthetic metrics, they remain significant at a 95% confidence level, as calculated by the confidence intervals. Additionally, subjective evaluations better demonstrate the differences in musicality. As shown in Tabel 2, removing DPO led to a significant decrease in subjective metrics: OVL dropped by 0.24 points, MEL by 0.22 points, showing a substantial reduction in musicality.
>
> Regarding the effectiveness of DPO in prompt consistency, the effect is less noticeable due to the limited capability of the MuQ model. We further conducted a MOS listening test to evaluate the alignment between  the prompt and the generated music. In this experiment, "LeVo" scored 3.175 ± 0.043, while "LeVo w/o DPO" scored 3.055 ± 0.057, indicating that DPO improves prompt consistency. (The value before the ± represents the mean score, while the value after the ± represents the error margin calculated at a 95% confidence level.)
>
> **Regarding Weakness 4: Reproducibility**
>
> To help readers reproduce our work, we have provided detailed information about dataset construction, model architecture, and training strategies in the paper and supplementatory material. Additionally, we commit to releasing the code and full checkpoints as soon as the paper is published.
>
> **Regarding Question 1: Claims of Lines 71-73**
>
> As mentioned in line 70 of the paper, we aimed to highlight the advantages brought by our parallel prediction strategy for mixed and dual-track tokens and the LeLM structure. As shown in Table 2, it can be observed that both YuE and SongGen, which predict dual-track tokens only, perform significantly worse than "LeVo w/o DPO" in overall quality (OVL), vocal melodic attractiveness (MEL), vocal-instrument harmony (HAM), and audio sound quality (AQ), confirming the advantage of our method in musicality, vocal-instrument harmony, and sound quality. When DPO is applied, compared to "LeVo w/o Dual-track" (which predicts only mixed tokens) and "LeVo w/o AR decoder" (which directly jointly trains to predict mixed and dual-track tokens), "LeVo" achieves better pefemance in musicality, vocal-instrument harmony, and sound quality.
>
> **Regarding Question 2: Clarity of Figure 2**
>
> We apologize for any confusion caused by Figure 2. The token sequences representing lyrics, text descriptions, and audio prompts are concatenated as a sequence and fed into the model, not summed. "Prompt Embedding" refers to the hidden states output by the last layer of the LeLM's language model after processing the concatenated token sequence (comprising lyrics, text descriptions, and audio prompts tokens). These states are passed to the AR Decoder to provide contextual information about the conditions. We will update the figure and method section to clarify this process and eliminate any potential confusion in the final version of the paper.
>
> **References:**
>
> [17] Lam, M. W. Y., Xing, Y., You, W., et al. "Analyzable chain-of-musical-thought prompting for high-fidelity music generation." *arXiv* *preprint* *arXiv:2503.19611*, 2025.
>
> [18] Yuan, R., Lin, H., Guo, S., et al. "YuE: Scaling open foundation models for long-form music generation." *arXiv* *preprint* *arXiv:2503.08638*, 2025.

---

> ### Comment · Reviewer_arNL · 2025-08-05
>
> Thanks for the detailed rebuttal.
>
> - **Significance Tests**: Thanks for the clarification. Please include the error bars to support the significance claims.
> - **Table 1**: Got it. Error bars would help here.
> - **Figure 2**: Sounds good.

---

### Official Review · Reviewer_YZXD · 2025-07-06

**Clarity:** 3
**Significance:** 2
**Originality:** 2
**Rating:** 4
**Confidence:** 3

**Summary:**

The paper proposed a song generation method based on lyrics, prompt audio, and text descriptions. The proposed method consists of 3 stages. First, dual-track (vocal and accompaniments) and mixed track based language model is trained to have a long-form characteristics of the song. Then, the dual-track tokens are injected into DiT like diffusion model for modeling high quality sound quality. Finally, to enhance the quality of the trained model's performance, the authors conducted DPO fine-tuning. At this stage, the authors collected thousands of human annotations over the generated music and lyrics, then with lyric alignment measurement algorithm and language music similarity measurement algorithm, the authors trained some reward model that predicts diverse thresholds of the quality. Then, the authors created large win-lose pairs using this reward model and fine-tuned the whole model to improve the performance of the model. In the experiments, the authors collected 2 million songs and generated according metadata using several algorithms. For example, they extracted vocals using separation model, and extracted lyric and timestamps through Whisper and wav2vec 2.0. And, musical section information was extracted using All-In-One model, and finally tags are extracted using Qwen2-Audio. Finally, they evaluated the generated music for both objective and subjective evaluations. In objective evaluation, they used FAD, PER, and text, audio alignment metrics with Meta Audiobox-Aesthetic scores. In subjective evaluation, the authors measured 6 MOS metrics. Overall, the proposed method showed superior performance than the public algorithms and competitive performance over industry models. Also, they reported the results of ablation studies, and it verified that the proposed dual-track model, DiT, DPO fine-tuning, and individual DPO methods are all effective, respectively. The paper is well-written and the experiments and evaluation seems correct.

**Questions:**

The concern lies on how the authors collected the training music data. They acquired 2 million songs, but it's not explained in the paper.

**Ethical Concerns:**

["Major Concern: Data privacy, copyright, and consent"]

**Final Justification:**

If the paper is accepted, I strongly urge the authors to keep this in mind and ensure that the reported results, training datasets, the released data collection codebase, and the released model checkpoints are all properly aligned.

**Limitations:**

-

**Paper Formatting Concerns:**

The letter size is decreased from the original letter size.

**Quality:**

3

**Strengths And Weaknesses:**

The use of dual-track and DPO fine-tuning steps seems novel and effective. Especially, how the authors constructed the training dataset and how the authors created reward model and used it for DPO fine-tuning are well described and will be beneficial for the future researchers.

---

> ### Author Rebuttal · Authors · 2025-07-31
>
> Thank you very much for your time and effort in reviewing our paper and for your recognition of our work. We hope our response below resolves your concerns and questions.
>
> **Regarding Question 1 and Ethical Concerns:** **Data Source**
>
> Regarding the training music data, we clarify this in the Supplementary Material, specifically in Appendix A (lines 6–7). We used a combination of licensed open-source datasets such as **DISCO-10M** [1*] and the **Million Song Dataset** [2*], along with a small portion of copyrighted proprietary data. This setup is similar to previous works in the field, including SongCreator [16], MusiCot [17], and YuE [18].
>
> **Regarding Formatting Concerns: Letter Size**
>
> We appreciate your valuable suggestion regarding the letter size. We will address this issue in the final version of the paper to ensure clarity.
>
> **References:**
>
> [1*] Lanzendörfer, L., Grötschla, F., Funke, E., et al. "Disco-10M: A large-scale music dataset." *Advances in Neural Information Processing Systems*, 2023, 36: 54451-54471.
>
> [2*] Bertin-Mahieux, T., Ellis, D. P. W., Whitman, B., & Lamere, P. "The Million Song Dataset." *In ISMIR*, volume 2, page 10, 2011.
>
> [16] Lei, S., Zhou, Y., Tang, B., et al. "Songcreator: Lyrics-based universal song generation." *Advances in Neural Information Processing Systems*, 37:80107–80140, 2024.
>
> [17] Lam, M. W. Y., Xing, Y., You, W., et al. "Analyzable chain-of-musical-thought prompting for high-fidelity music generation." *arXiv* *preprint* *arXiv:2503.19611*, 2025.
>
> [18] Yuan, R., Lin, H., Guo, S., et al. "YuE: Scaling open foundation models for long-form music generation." *arXiv* *preprint* *arXiv:2503.08638*, 2025.
>
> Note: [N] denotes the reference in the paper and [N*] represents the reference in the Appendix.

---

> > ### Comment · Reviewer_YZXD · 2025-08-05
> >
> > I sincerely hope that the authors have been very clear regarding the training data (as answered) and that the reported model performance in the paper was fairly obtained using the stated datasets. If the paper is accepted, I strongly urge the authors to keep this in mind and ensure that the reported results, training datasets, the released data collection codebase, and the released model checkpoints are all properly aligned.

---

### Note · Authors · 2025-08-14

We extend our sincere gratitude to the Area Chairs and all reviewers for dedicating their valuable time to reviewing our paper and providing constructive feedback. We believe these comments are crucial for enhancing the overall quality of this paper.

We are delighted that the reviewers appreciate our paper from various perspectives, including the novelty and effectiveness of the model architecture and multi-preference alignment method [YZXD, arNL, XdTb, 7iGT], clear and well-structured writing [YZXD，arNL，7iGT], comprehensive experiments and ablation studies [YZXD, arNL, XdTb, 7iGT], remarkable music genration performance [YZXD，arNL，XdTb], and the effective use of pretrained models for data preparation and evaluation [7iGT]. These positive assessments truly motivate us.

We have provided detailed, point-by-point responses to each reviewer to address their concerns. The main revisions in our response are summarized as follows:

**More comprehensive experiments and model descriptions**

- We have included 95% confidence intervals (CI95) for both the objective and subjective experiments.
- We have added a “LeVo w/o DPO” row in Table 1.
- We have updated Figure 2 and Section 3.2 to more clearly illustrate how lyrics, text descriptions, and audio prompts are fed into the model, and to clarify the meaning of prompt embeddings.
- We have added experiments to explore the impact of pretrained model quality on overall performance.
- We have conducted an analysis on the correlation between objective metrics and subjective evaluation results to explore the limitations of each objective metric.
- We have conducted an analysis to investigate the contributions of each module(LeLM, Codec, and VAE) to the final audio quality.

**On reproducibility**

- We commit to releasing the full codebase and checkpoints to support reproducibility.

**On data ethics**

- We have clarified the full details of the datasets used. Apart from these datasets, we confirm that no additional music was scraped from the internet and no copyrighted commercial content was used without authorization.
- We have further analyzed the ethical implications and risks of model memorization. Our results show that using identical lyrics is unlikely to cause training data leakage.

Finally, we thank the reviewers for their recognition of our paper and responses, and we hope our clarifications have addressed their concerns.

---

### Decision · Program_Chairs · 2025-09-17

**Decision:**

Accept (poster)

**Comment:**

This paper introduces LeVo, a conditional music generation model trained through a three-stage pipeline: (1) autoregressive pretraining on mixed audio tokens, (2) fine-tuning on separated vocal and accompaniment tokens to improve fidelity, and (3) preference alignment via direct preference optimization (DPO) to enhance lyric alignment and consistency. The proposed method addresses key limitations of prior work, such as low fidelity, vocal–instrument mismatch, and noisy training data.

All reviewers find the submission valuable, novel, and well-executed, with strong empirical results. In the rebuttal, the authors satisfactorily addressed concerns regarding missing results (e.g., error bars, baselines without DPO), reproducibility, and datasets details. I strongly encourage the authors to incorporate these clarifications and maintain their commitment to open-sourcing in the final version.

In light of these strengths, I recommend acceptance.